# Phenological Changes of Woody Plants in the Southern and Northern Regions of Nanling Mountains and Their Relationship with Climatic Factors

**Guangxu Liu [1],\*, Aicun Xiang [1], Zhiwei Wan [1], Haihui Lv [1,2], Haolong Liu [3,4], Zhen Hu [1] and Lili Chen [1]**

1   School of Geography and Environmental Engineering, Gannan Normal University, Ganzhou 341000, China; xiangaicun@gnnu.edu.cn (A.X.); wanzw.09b@igsnrr.ac.cn (Z.W.); qzyxyangyuhui@foxmail.com (H.L.); huzhen@gnnu.edu.cn (Z.H.); 211401063@gnnu.edu.cn (L.C.)
2   Chinese Language and Culture College, Huaqiao University, Xiamen 361021, China
3   Institute of Geographic Sciences and Natural Resources Research, Chinese Academy of Sciences, Beijing 100101, China
4   Key Laboratory of Land Surface Pattern and Simulation, Chinese Academy of Sciences, Beijing 100101, China
\*   Correspondence: liuguangxu@gnnu.edu.cn

**Abstract:** In addressing the challenges posed by the implications of climate change, understanding the phenological variations of woody plants has become a pivotal research topic. This research centers on the phenological shifts of woody plants and their connections with climatic factors in the southern and northern regions of the Nanling Mountains, which serve as the boundary between the north subtropical climate zone and the south subtropical climate zone in South China. The data were gathered through extensive manual observations conducted at four plant phenology observation stations (Ganxian, Foshan, Guilin, and Changsha) spanning different periods from 1963 to 2008. The study scrutinized four widely distributed woody plant species in the research area, specifically *C. mollissima*, *P. fortunei*, *M. azedarach*, and *M. grandiflora*. The analytical methods utilized were linear trend estimation and Pearson correlation coefficient analyses. The principal findings were as follows: (i) over the past several decades, the phenological stages of woody plants in the southern region consistently preceded those in the northern region with variations ranging from 2 to 38 days; (ii) an advancing trend of 0.1 to 2 days per decade was discerned in the phenological stages of all woody plants in the southern region; (iii) within the same geographic region, distinct species exhibited varying sensitivities to climatic factors, and *M. azedarach* demonstrated a particularly high sensitivity to climate fluctuations affecting phenological stages; and (iv) different climatic factors exerted distinct influences on individual plant species. Notably, temperature emerged as the primary driver of phenological changes, which was supported by a significant negative correlation between the phenological stages of the studied plants and spring temperatures. This study contributes to our understanding of the effects of climate change on plant phenology and offers valuable insights to guide ecological conservation and management strategies within the region.

**Keywords:** phenological stages; woody plants; climate change; plant-climate interactions; Nanling Mountains

## 1. Introduction

Climate change is a significant challenge that is currently faced globally and exerts broad and profound impacts on natural ecosystems and human societies [1]. With rising temperatures, altered precipitation patterns, and changes in other climatic factors, important life cycle events of plants, including growth, flowering, and fruit ripening, have exhibited significant changes [2]. These changes have had substantial and far-reaching effects on plant phenology, which refers to the seasonal patterns of growth and development. It is important to use plant phenology as a sensitive indicator of climate change

because global warming has a substantial impact on plant phenology [3]. For instance, Partanen et al. (1998) found that rising temperatures altering photoperiod and temperature conditions may impede the premature onset of growth in Norway spruce (*Picea abies (L.) Karst.*) in response to climate change [4]. Walther et al. (2002) and Cleland et al. (2007) found that climate change had significant effects on the timing and spatial distribution of phenological events in global ecosystems [5,6]. Vitasse et al. (2009) uncovered that the leaf phenology of European interpopulation tree species responded to temperature differences, providing important clues for predicting changes in plant phenology [7]. Menzel (2003) observed a correlation between plant phenological changes and temperature and the North Atlantic Oscillation (NAO) [8]. In a recent study, Vitasse et al. (2022) observed that in the 19th century, flowering and budburst dates remained stable. However, in the first half of the 20th century, spring phenological events in Switzerland and Japan began to advance, which was consistent with increasing temperatures. The strongest advancement in spring phenology was found from the mid-1980s onward. Over the 36 years from 1985 to 2020, spring phenology advanced by 6 days (China) to 30 days (Switzerland) compared with the period before 1950. This observation aligns with the accelerating warming trends observed across study sites and the Northern Hemisphere [9].

While the impact of climate change on phenology is widely acknowledged, the response of phenology to climate change is notably intricate [10]. Wenden et al. (2022) discovered that with climate warming in colder latitudes, the forcing period of European beech (*Fagus sylvatica* L.) and pedunculate oak (*Quercus robur* L.) has extended over recent decades. However, in warmer latitudes, this period has shortened for both species, with a more pronounced shift observed in beech [11]. Piao et al. (2019) indicated that the complex interactions between multiple driving factors complicated the modeling and prediction of plant phenological changes [12]. Asse et al. (2018) revealed the impact of long-term climate warming on plant abundance. They found that it led to an earlier onset of budburst and flowering in herbaceous plants. Meanwhile, sedge plant abundance decreased under the same climatic conditions. Notably, these changes occurred without affecting aboveground net primary productivity [13]. Liu et al. (2022) analyzed 88 published studies and found mismatched effects of climate warming on aboveground and belowground plant phenology. In herbaceous plants, climate warming advanced the start and end dates of aboveground growing seasons without affecting belowground phenology. For woody plants, climate warming did not affect aboveground phenology but extended their belowground growing season [14]. These studies show that plant phenology could serve as a critical component of climate change fingerprints, and it has the potential to detect climate change through phenological responses [15]. However, the relationship between climate change and phenological shifts is exceedingly complex. Further exploration of such complexity in different areas could enhance our understanding of plant adaptation strategies to climate variations.

As a region of significant biodiversity and ecological importance, the Nanling Mountains in China present an ideal landscape for investigating the phenological responses due to their distinctive geographical and climatic conditions. The southern region is characterized by a tropical or southern subtropical climate, with higher temperatures, hot and humid summers, and warm winters. By contrast, the northern region falls within the north subtropical climate zone, featuring lower temperatures, cooler summers, cold winters, and lower precipitation. Containing diverse vegetation types, including forests, grasslands, and wetlands, the Nanling Mountains harbor a rich array of plant species that likely exhibit varying responses to climate change. Cao et al. (2012) found that 39 tropical tree species introduced artificially along the northern border of the Nanling Mountains in Ganzhou City were able to grow successfully; this was primarily attributed to the ongoing climate warming in the area [16]. Yuan et al. (2017) found that climate warming increased the vulnerability of subtropical evergreen broad-leaved forests, including those in the Nanling Mountains, in terms of NPP [17]. Li et al. (2019) predicted a significant upward shift due to future climate warming to the upper forest line in the mountainous areas of the Nanling Mountains, which poses a threat to high-altitude tree species such as Chinese fir

(*Cunninghamia lanceolata* (Lamb.) Hook.) [18]. Peng et al. (2021) found that temperature variations had a significant impact on the end of the growing season (EOS) of mountainous vegetation in the Xiangjiang River Basin, which is in the northern Nanling Mountains [19]. These studies, using remote sensing data or modeling approaches, have contributed to our understanding of the complex changes in phenology in response to climate change. However, direct observational data validating the differences in phenological responses to climate between the southern and northern regions of the Nanling Mountains are missing.

This study narrows its focus by comparative analysis, specifically examining the differential phenological responses to climate changes in the northern and southern regions of the Nanling Mountains with observation data, with the hypothesis that the differential impacts of climate change on the phenology in both regions will reveal distinct variation patterns in response to climatic factors. The research, initiated by the Chinese Academy of Sciences in the 1960s, established phenological observations in Guilin, Changsha, Ganxian, and Foshan on both sides of the Nanling Mountains. Over a span of more than 40 years, these observations have accumulated valuable phenological data. This research, building upon previous research and datasets, delves into the phenological changes of *Castanea mollissima* Bl. (*C. mollissima*), *Melia azedarach* L. (*M. azedarach*), *Magnolia grandiflora* Linn. (*M. grandiflora*), and *Paulowinia fortune* (seem.) Hemsl. (*P. fortunei*). The results contribute to advancing our understanding of the complex impact of climate change on the local ecosystem. The findings provide insights into plant adaptation, population dynamics, and ecosystem stability in response to climate change. Moreover, the study offers valuable reference information for ecological conservation and climate change adaptation management.

## 2. Study Area

The geographical location of the core area of the Nanling Mountains is approximately between 24°00′ N−26°30′ N and 109°30′ E−116°45′ E. It spans a length of approximately six hundred kilometers from east to west and a width of about two hundred kilometers from north to south. The region is located at the intersection of four provinces: Hunan, Jiangxi, Guangdong, and Guangxi, serving as a watershed between the Yangtze River and Pearl River systems. The topography of the Nanling Mountains is characterized by mountainous hills, including five prominent mountain ranges: Dayuling, Qitianling, Mengzhuling, Duopangling, and Yuechengling (Figure 1). The latitude range was extended by 2° in the north–south direction, setting the northern and southern boundaries of the study area as 22°00′ N–28°30′ N. The dividing line between the northern and southern regions was set at 25°30′ N. The observation sites of Ganzhou and Changsha, which is from the China Phenological Observation Network, were categorized as the northern region of the Nanling Mountains, while Guilin and Foshan were categorized as the southern regions.

The Nanling Mountains serve as the boundary between the central subtropical zone and the southern subtropical zone in China. Quantitative analysis of regional characteristics in the Nanling Mountains, considering the spatial heterogeneity, reveals significant spatial differentiation between the two sides of the mountain range [20]. The northern part of the Nanling Mountains has an annual average temperature ranging from 16 to 20 °C, average temperatures during the coldest month ranging from 3 to 8 °C, and the annual precipitation ranging from 1200 to 1600 mm. It has the climatic characteristics of a rainy spring and hot summer. The southern part has a tropical and south-subtropical climate with an annual average temperature ranging from 21 to 23 °C, average temperatures during the coldest month ranging from 13 to 18 °C, and the annual precipitation ranging from 1500 to 2000 mm. Due to the topographic relief and the distance from the sea, the eastern part of the Nanling Mountains has a humid climate with a large amount of precipitation, an annual average precipitation exceeding 2000 mm over 140 to 200 days. The western part of the Nanling Mountains has a dry climate with an annual average precipitation of 1000 to 1500 mm over 100 to 140 days. Overall, the southern slope exhibits higher temperatures and humidity levels compared to the northern slope [21], markedly impacting species distribution.

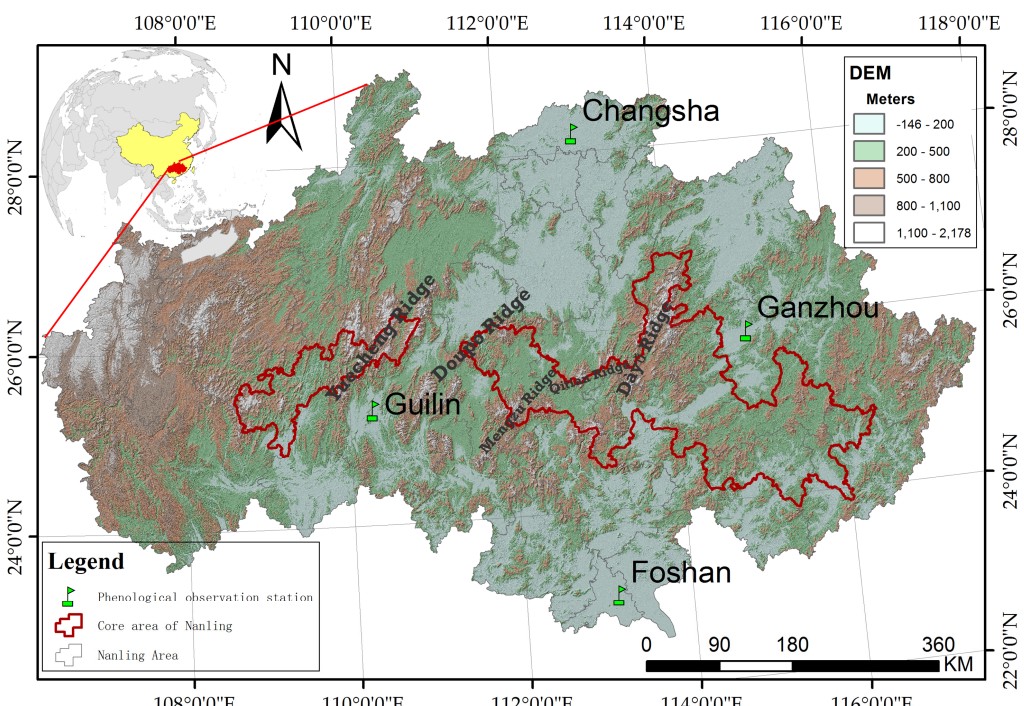

**Figure 1.** Location of Nanling Mountains in China. The selected stations are denoted by green flags in the map, with Guilin and Foshan located in the southern part of the study area, and Changsha and Ganxian are situated in the northern region of the research area. Digital Elevation Model (DEM) of the study area created by ASTER GDEM (Advanced Spaceborne Thermal Emission and Reflection Radiometer Global Digital Elevation Model) at 30 m resolution, collected from http://mapgl.com/shareData/ (accessed on 13 June 2023). The map coordinates are in the Krasovsky_1940 reference system with Albers projection.

Situated in an ecotone influenced by climate, the natural landscape of the Nanling Mountains exhibits a transitional zone where characteristics of both subtropical and tropical regions coexist. Dominated by evergreen broad-leaved forests, this region harbors a significant reservoir of ancient, relict, primitive, and endemic plant species, making it a vital component of the modern Chinese plant floristic province [22]. Notably, *C. mollissima*, *P. fortunei*, *M. azedarach*, and *M. grandiflora* exemplify key species from Fagaceae, Paulowniaceae, Meliaceae and Magnoliaceae, respectively. *C. mollissima*, also known as chestnut, is characterized by its flowering period from April to June and fruit maturation in September (Figure 2), and it is widely distributed from lowlands to elevations of 2800 m. *P. fortunei*, which is commonly referred to as the princess tree or phoenix tree, blooms from March to April and bears fruit from October to December. It thrives in low-lying wild areas, roadsides, and sparse forests, and it is widely distributed in tropical and subtropical regions of Asia. *M. azedarach*, also called Chinaberry or Chinese Mahogany, flowers from April to May and produces fruit from November to December. It thrives in moist and fertile soils. *M. grandiflora*, specifically Magnolia grandiflora, also known as southern magnolia or bull bay, blooms from May to June and bears fruit in August and September. With a straight trunk and rapid growth, *M. grandiflora* is adaptable and primarily distributed in low-elevation slopes, forests, valleys, and wastelands. The four selected woody plant species are extensively distributed in the Nanling Mountains, which is renowned for its representative nature. Moreover, these four species have long-term and relatively complete records with high data accuracy across the four phenological observation sites in Ganxian, Changsha, Foshan, and Guilin.

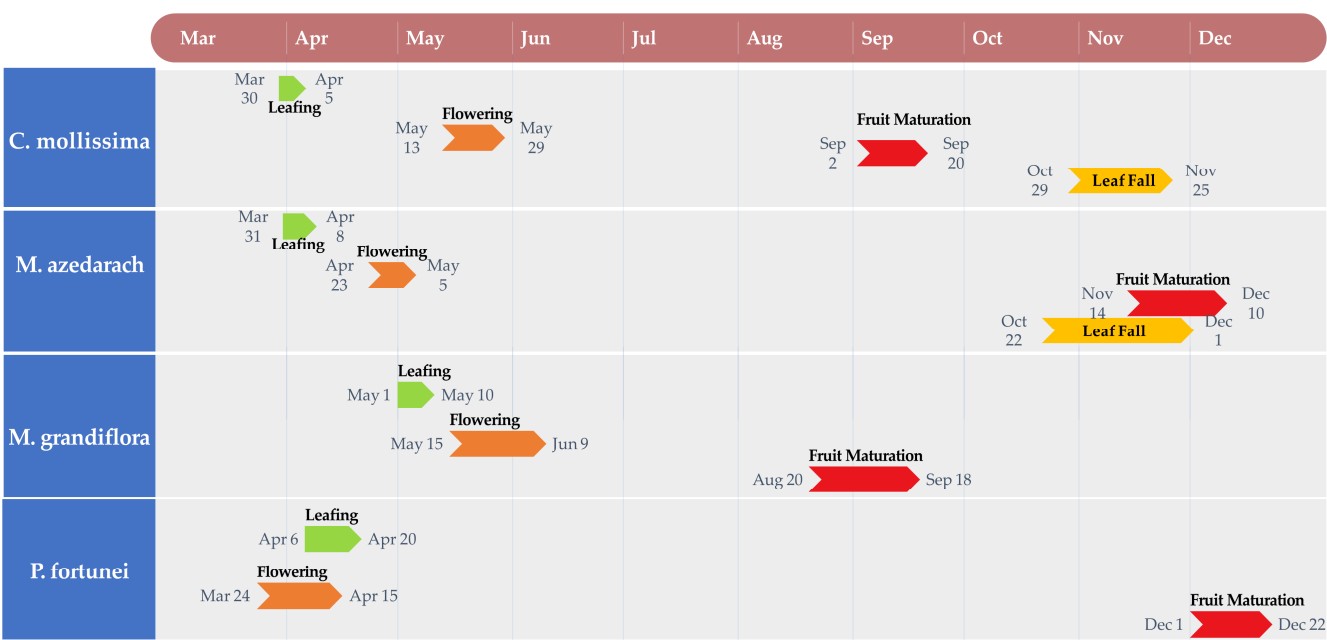

**Figure 2.** Phenological chart of the selected wood species. It appears the median dates of phenological stages in the four stations observed during 1963–2008.

## 3. Data and Methods

### 3.1. Data

The phenological observation data used in this study were from the China Phenological Observation Network and obtained from the National Earth System Science Data Center, National Science & Technology Infrastructure of China (http://www.geodata.cn, accessed on 29 January 2021). The observed entities include 35 commonly observed plant species (observed in all the stations), 127 locally observed plant species (not observed in all the stations), 12 animal species, 4 crops, and 12 meteorological and hydrological phenomena. The data are collected in accordance with the observation requirements specified in the *China Phenological Observation Guidelines.* Several measures were implemented to ensure the quality of observations, including the following: ① Observers are personnel who have undergone rigorous professional training. ② Observed entities are fixed at specific locations, including trees and plants, with most observation sites being flat and open areas maintained as fixed observation points over multiple years. ③ Selected tree species are in normal development with a minimum of 3 years of flowering and fruiting. ④ Observations are recorded onsite with daily accuracy, near the observation locations. Four woody plant species (*C. mollissima*, *P. fortunei*, *M. azedarach*, and *M. grandiflora*) were selected from four observations (Guilin, Changsha, Ganxian, and Foshan) located on both sides of the Nanling Mountains. The key phenological indicators, namely the dates of flowering start and flowering end, were chosen to analyze the characteristics of phenological changes and their responses to climate change. According to *China Phenological Observation Guidelines*, the flowering start is defined as the date when at least three locations observe complete blooming. For selected trees of the same species, the start is recorded when at least 50% of the trees have three or more petals with flowers fully open simultaneously. The flowering end is defined as the date when fewer than 5% of the observed trees have flowers remaining [23]. To ensure data continuity and completeness, the period from 1963 to 2008 was selected as the study period. In the selected period, all four woody plant species had missing values due to factors such as insufficient funding, station closures, and interruptions in observational records (Table 1). To prevent data distortion, missing values were allocated to unobserved years. The dates were converted using the Julian day method. The Julian day is a numerical representation of dates proposed by the Egyptian astronomer Julius Scaliger in the 16th century. It measures the count of days from the

initial moment of the Gregorian calendar, which began at noon on January 1, 4713 BC. The use of Julian days facilitates numerical calculations and the processing of dates [24]. Meteorological data, including monthly average temperature, monthly precipitation, and sunshine duration, were obtained from the National Meteorological Information Center–China Meteorological Data Network (www.cma.cn, accessed on 22 February 2021) and selected from the meteorological stations closest to the four phenological observation sites. In this study, correlation analysis was employed to examine the phenological response characteristics of representative species in the face of climate change. Specifically, the study employed non-parametric correlation analysis, focusing on Spearman's rank correlation test, to explore the correlation between different parameters. Linear regression, employing the least squares method, was applied to analyze the relationship between meteorological factors and phenological stages.

**Table 1.** Years with missing observation records at stations. "m a.s.l" is the abbreviation of "meters above sea level".

| Stations | Location | | Elevation (m a.s.l) | Years with Missing Observation Records | | | |
|---|---|---|---|---|---|---|---|
| | Latitude (N) | Longitude (E) | | *C. mollissima* | *M. azedarach* | *M. grandiflora* | *P. fortunei* |
| Changsha | 28.2 | 112.9 | 82 | 1963–1964, 1973–1974 | - | 1964–1974, 2003–2005 | 1963–2004 |
| Ganzhou | 25.9 | 114.9 | 101 | 1963–1976, 1991, 1996–2008 | 1963–1976, 1991, 1996–2008 | 1963–1988, 1996–2008 | 1963–1976, 1990–1991, 1996–2008 |
| Guilin | 25.3 | 110.3 | 159 | 1963, 1968–1972, 1980, 1989, 1992, 1994–2002 | 1963, 1968–1972, 1992, 1994–2004 | 1963–1978, 1994–2002, 2007 | 1963–1976, 1990–1991, 1994–2002 |
| Foshan | 23 | 113.1 | 17 | 1963–1964, 1968, 1981–1983, 1988–1989, 1991–2002, 2003–2008 | 1988, 1991–2002 | 1963–2002 | 1963–1967, 1969–1972, 1972–1978, 1991–2008 |

*3.2. Methods*

Spearman's rank correlation is a statistical technique that transforms the original data into ranks, ranging from the smallest to the largest. By computing the correlation between the ranks of the two variables, rather than using the original numerical values directly, it provides a measure of the relationship between the variables. This method is particularly useful for assessing nonlinear relationships and analyzing data that do not follow a normal distribution [25]. The formula for calculating Spearman's rank correlation coefficient (rho) is as follows:

$$\rho = 1 - \frac{6 \sum d^2}{n * (n^2 - 1)} \tag{1}$$

where $\rho$ represents the Spearman rank correlation coefficient, $\Sigma d^2$ represents the sum of squared rank differences, and $n$ denotes the sample size.

The least squares method is a commonly used statistical approach employed in linear regression fitting. Its fundamental principle is to minimize the sum of squared residuals between the observed values and the fitted model, thereby determining the optimal fitting line. This method aims to minimize the overall distance between the observed data points and the line, resulting in the smallest possible differences between the observed values and the fitted values [26]. The computational formula is as follows:

$$\beta = \left( X^{\mathrm{T}} * X \right)^{-1} * X^{\mathrm{T}} * Y \tag{2}$$

where $\beta$ represents the vector of regression coefficients, $X$ represents the matrix of independent variables, $Y$ represents the vector of dependent variables, and $(^{\mathrm{T}})$ denotes the transpose of a matrix.

The comparative study of phenological variations between the northern and southern regions of the Nanling Mountains was conducted from four perspectives and two dimensions. The four perspectives included a comparison between the eastern and western parts of the northern region, a comparison between the eastern and western parts of the southern region, a comparison between the northern and southern regions of the eastern part of the Nanling Mountains, and a comparison between the northern and southern regions of the western part of the Nanling Mountains. The two dimensions encompassed a comparison of phenological dates among plants and an analysis of the correlation between key phenological stages and meteorological observation points.

## 4. Results

### 4.1. Characteristics of the Plant Phenology

4.1.1. Characteristics of the Plant Phenology in the Northern Region

The observational data from Ganxian, located at 114.9° east longitude (E) and 25.9° north latitude (N) in the Nanling Mountains, covered a period of 32 years, spanning from 1964 to 1996. The analysis of this data revealed that the flowering start dates of four woody plant species, namely *C. mollissima*, *M. azedarach*, *M. grandiflora*, and *P. fortunei*, were primarily observed between early February and late May (15 February to 31 May) and the flowering end dates were primarily observed between early March and early June (6 March to 10 June). Notably, there was a variation of 25 to 55 days between the earliest and latest dates of flowering start and flowering end, indicating unstable changes. Among them, *P. fortunei* exhibited the most pronounced fluctuations.

At Changsha (112.9° E, 28.2° N), valid observational data were available for a period of 44 years: from 1964 to 2008. The flowering start dates of *C. mollissima*, *M. azedarach*, *M. grandiflora*, and *P. fortunei* in Changsha were concentrated between mid-March and early June (15 March to 5 June), and the flowering end dates were concentrated between late March and late June (30 March to 30 June). Analysis revealed that there was a notable disparity between the earliest and latest flowering start dates of *C. mollissima* in Changsha, whereas the difference for the flowering end dates was smaller, indicating a higher variability of the flowering start. Moreover, when comparing with those in Ganxian, the phenological fluctuations of *P. fortunei* in the Changsha region were comparatively smaller, and the average dates of flowering start and end were substantially late compared to those at Ganxian.

4.1.2. Characteristics of Plant Phenology in the Southern Region

Based on observational data from Foshan, which is located at 113.1° E and 23° N along the southern slope of the Nanling Mountains, the flowering start dates of four woody plant species, namely *C. mollissima*, *M. azedarach*, *M. grandiflora*, and *P. fortunei*, were concentrated between mid-February and late May (12 February to 20 May). The flowering end dates occurred from mid-March to late May (17 March to 30 May). Among these species, except for *M. grandiflora*, there was a variation in the timing of both the earliest and latest flowering start as well as the flowering end in Foshan. Compared to Ganxian and Changsha in the northern region of the Nanling Mountains, the four plant species at Foshan exhibited an earlier flowering start.

At Guilin (110.3° E, 25.3° N), the flowering start dates of *C. mollissima*, *M. azedarach*, *M. grandiflora*, and *P. fortunei* were concentrated between early February and early June (8 February to 9 June) with the flowering end dates occurring from mid-March to mid-June (18 March to 20 June). Analyzing the phenological changes, it was found that there were no differences in the timing of flowering onset and flowering end among the four woody plant species. However, *P. fortunei* showed greater fluctuations in both phenological stages.

4.1.3. Comparison of the Phenological Phases between the Northern and Southern Regions

Comparison of the considering phenological characteristics of four woody plant species at Ganxian and Foshan revealed that the phenological stages (flowering start and

flowering end) at Ganxian occurred earlier than those in the southern region (Table 2). A consistent pattern was observed between Changsha and Guilin, except for the flowering end of *P. fortunei*, which exhibited a similar average date in both the southern and northern regions. However, the phenological stages of the other woody plant species showed an earlier occurrence in the southern region compared to the northern region (Table 3). Overall, the phenological stages in the southern region of the Nanling Mountains consistently preceded those in the northern region (Tables 2 and 3).

**Table 2.** Comparison of plant phenology between Ganxian and Foshan (MM/DD).

| Phenological Stages | Species | Ganxian Average Date | Foshan Average Date | Differences in Days | Comparison Results |
|---|---|---|---|---|---|
| Flowering start | *C. mollissima* | 5/10 | 4/29 | 11 | Earlier in the south |
| | *M. azedarach* | 4/13 | 3/21 | 23 | Earlier in the south |
| | *M. grandiflora* | 5/16 | 4/20 | 26 | Earlier in the south |
| | *P. fortunei* | 3/13 | 2/28 | 14 | Earlier in the south |
| Flowering end | *C. mollissima* | 5/27 | 5/13 | 14 | Earlier in the south |
| | *M. azedarach* | 4/19 | 4/10 | 9 | Earlier in the south |
| | *M. grandiflora* | 6/7 | 5/17 | 21 | Earlier in the south |
| | *P. fortunei* | 4/17 | 4/14 | 3 | Earlier in the south |

**Table 3.** Comparison of plant phenology between Changsha and Guilin (MM/DD).

| Phenological Stages | Species | Changsha Average Date | Foshan Average Date | Differences in Days | Comparison Results |
|---|---|---|---|---|---|
| Flowering start | *C. mollissima* | 5/17 | 4/29 | 18 | Earlier in the south |
| | *M. azedarach* | 4/28 | 3/21 | 38 | Earlier in the south |
| | *M. grandiflora* | 5/14 | 4/20 | 24 | Earlier in the south |
| | *P. fortunei* | 3/25 | 2/28 | 26 | Earlier in the south |
| Flowering end | *C. mollissima* | 6/3 | 5/13 | 21 | Earlier in the south |
| | *M. azedarach* | 5/11 | 4/10 | 31 | Earlier in the south |
| | *M. grandiflora* | 6/14 | 5/17 | 28 | Earlier in the south |
| | *P. fortunei* | 4/12 | 4/14 | 2 | Almost the same |

*4.2. Interannual Variations of the Plant Phenology*

4.2.1. Phenological Stage Variations in the Northern Region

Figure 3 depicts the interannual variations of the flowering start and flowering end dates of the four plants at Ganxian. The *C. mollissima* and *M. azedarach* flowering start in the upper panel of Figure 3 exhibited substantial fluctuations during the mid to late 1980s (1986–1988). The *P. fortunei* flowering start showed noticeable variability during the mid-1980s (1984–1986). The bottom panel of Figure 3 illustrates a similar pattern in the variations of flowering end with *C. mollissima* and *M. azedarach* showing smoother fluctuations compared to their flowering start counterparts, while *P. fortunei* and *M. grandiflora* exhibited greater fluctuations in flowering start compared to flowering end.

Figure 4 shows the interannual variations of the flowering start and flowering end of the four plant species at Changsha. The flowering start at Changsha underwent two phases of change. In the first phase, from the mid-1970s to the mid-1980s (1974–1985), *M. azedarach* exhibited a delayed flowering start. In the second phase, from 1985 to 2008, the flowering start of all four species gradually advanced with fluctuating changes. Prior to 1985, the flowering end of *M. azedarach* remained stable, but after 1985, the flowering end of all species gradually advanced with fluctuating changes. Comparing the two phenological stages, both the flowering start and flowering end showed a decreasing trend from 1991 to 2007. This indicates a slow advancement of the flowering start and flowering end for the four plant species during the period of 1991–2007.

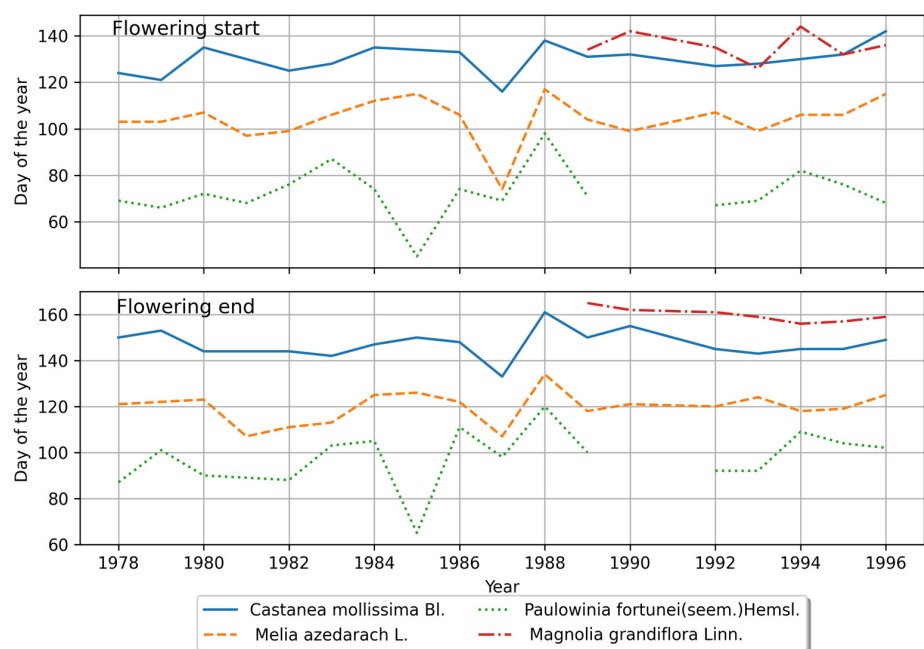

**Figure 3.** The interannual variations of the flowering start and flowering end dates of the woody plants at Ganxian. The discontinuous segments in the line graph represent unobserved years. The observational data for *M. grandiflora* span less than 20 years, and the trend changes depicted in Figure 3 do not reach statistical significance. The inclusion of the line is solely for comparative reference, as is the case in subsequent figures.

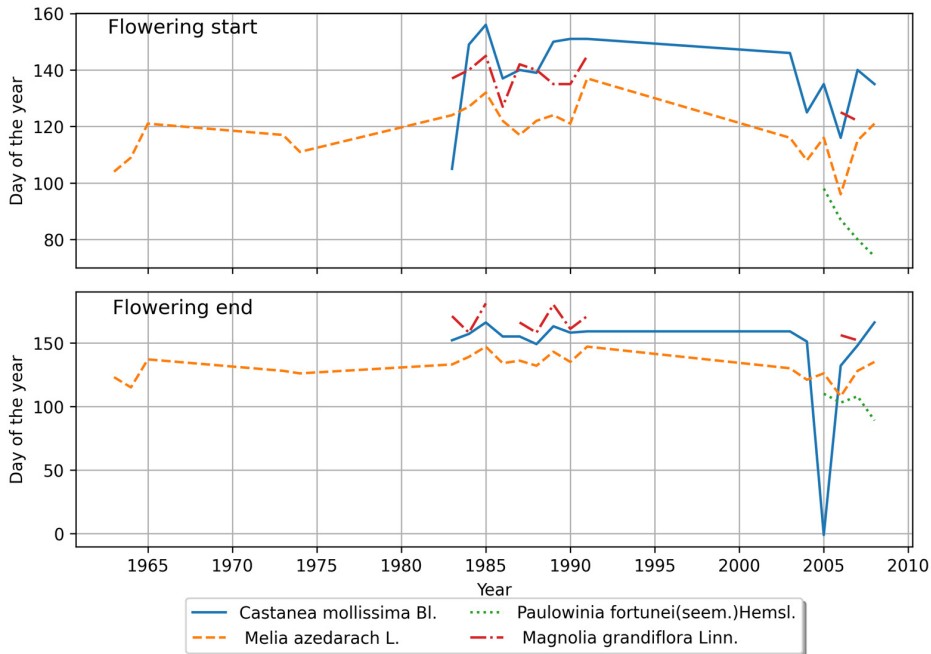

**Figure 4.** The interannual variations of the flowering start and flowering end dates of the woody plants at Changsha.

Table 4 shows the results of a linear regression analysis conducted on the interannual variations in the flowering start and end at Ganxian. The flowering start of *C. mollissima*, *M. azedarach*, and *P. fortunei* exhibited respective delays of 1.5, 2.1, 2.1, and 2.1 days per decade. However, the trends of flowering end for *C. mollissima*, *M. azedarach*, and *P. fortunei* were more complex. The delays in flowering end per 10 years were less than one day for *C. mollissima* and *M. azedarach*, while *P. fortunei* exhibited delays of 6.7 days, respectively,

per 10 years. Comparatively, *P. fortunei* showed larger variations in phenological stages, with the flowering end exhibiting more changes compared to the flowering start. Both *C. mollissima* and *M. azedarach* exhibited delays in flowering start and flowering end, while the changes in flowering end were stable.

**Table 4.** Linear regression analysis of interannual variations in the flowering start and flowering end dates at Ganxian.

| Phenological Stages | Species | Linear Regression Equation (Where x Represents Days of the Year) |
|---|---|---|
| Flowering start | *C. mollissima* | y = 0.1501x − 167.85 |
| | *M. azedarach* | y = 0.2051x − 303.38 |
| | *M. grandiflora* | y = 0.2103x − 282.76 |
| | *P. fortunei* | y = 0.2119x + 348.57 |
| Flowering end | *C. mollissima* | y = 0.0339x + 79.65 |
| | *M. azedarach* | y = 0.0684x − 16 |
| | *M. grandiflora* | y = 0.6948x − 1225.19 |
| | *P. fortunei* | y = 0.6712x − 1236.03 |

The fitting results for the Changsha site are presented in Table 5. Due to a shorter observation period and limited data, it was not possible to establish a fitting model for *M. grandiflora*. Based on the fitting outcomes, the flowering start and flowering end dates of *C. mollissima* and *M. azedarach* exhibited an advancing trend. Specifically, the flowering start dates of *C. mollissima* and *M. azedarach* advanced by less than one day on average per 10 years. The flowering end of *C. mollissima* advanced by an average of 9 days per 10 years, and *M. grandiflora* advanced by an average of 6.8 days per 10 years, while the flowering end of *M. azedarach* showed minor changes. When compared to the phenological stages at the Ganxian in the eastern Nanling Mountains, the phenological changes at the Changsha site exhibited an opposite trend. The phenological stages at the Ganxian site were predominantly delayed, whereas those at the Changsha site were mostly advanced.

**Table 5.** Linear regression analysis of interannual variations in the flowering start and flowering end dates at Changsha.

| Phenological Stages | Species | Linear Regression Equation (Where x Represents Days of the Year) |
|---|---|---|
| Flowering start | *C. mollissima* | y = −0.0626x + 262.28 |
| | *M. azedarach* | y = −0.0136x + 143.866 |
| | *M. grandiflora* | y = −0.7758x + 1681.13 |
| | *P. fortunei* | - |
| Flowering end | *C. mollissima* | y = −0.9024x + 1942.44 |
| | *M. azedarach* | y = −0.034x + 190.84 |
| | *M. grandiflora* | y = −0.6773x + 1513.85 |
| | *P. fortunei* | - |

4.2.2. Phenological Stage Variations in the Southern Region

Figure 5 presents the interannual variations in the flowering start and flowering end of the four woody plant species in the southern region at Foshan. Due to the limited duration of observations, data for *M. grandiflora* were insufficient for trend analysis. The flowering start of *C. mollissima* at the Foshan station in the upper panel of Figure 4 exhibited a gradual and slightly delayed trend. In contrast, *M. azedarach* showed minor fluctuations in the flowering start, displaying an advancing trend. The daily variation of *M. azedarach* flowering start showed greater fluctuations with values oscillating around 40 days. As depicted in the bottom panel of Figure 5, the flowering end of *C. mollissima* at the Foshan station exhibited a delayed trend. *P. fortunei*, on the other hand, displayed minor fluctuations with a slight

advancing trend in the flowering end. Furthermore, *M. azedarach* experienced a delay in the flowering end around 1976, which was followed by a subsequent overall advancing trend.

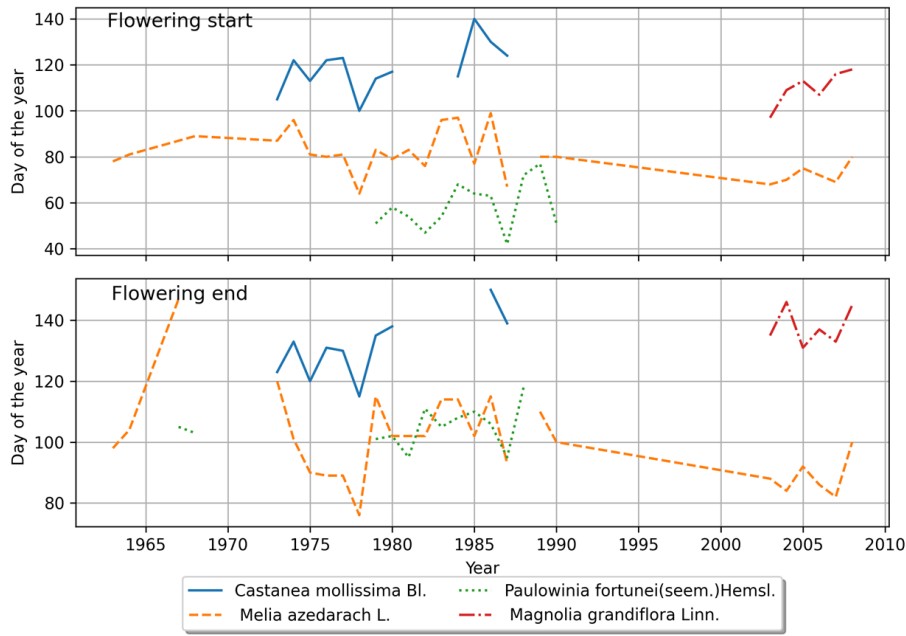

**Figure 5.** Interannual variations of the flowering start and flowering end dates of the woody plants at Foshan.

Table 6 presents the linear regression results for the variables of year and phenological stages at Foshan. The average delay in the flowering start of *C. mollissima* and *P. fortunei* at Foshan station was 6.3 days and less than 1 day, respectively, while *M. azedarach* exhibited an average advancement of 2.8 days per 10 years. The flowering end of *C. mollissima* and *P. fortunei* showed an average delay of 7.5 days and 3.1 days per 10 years, respectively, whereas *M. azedarach* exhibited an average advancement of 4.8 days per 10 years in its flowering end. Overall, among the three plant species at Foshan station, *C. mollissima* exhibited the greatest variation in phenological stages, and the changes in flowering end were more pronounced than those in flowering start.

**Table 6.** Linear regression analysis of interannual variations in the flowering start and flowering end dates at Foshan.

| Phenological Stages | Species | Linear Regression Equation (Where x Represents Days of the Year) |
|---|---|---|
| Flowering start | *C. mollissima* | y = 0.6253x − 1119.093 |
| | *M. azedarach* | y = −0.2795x + 634.98 |
| | *M. grandiflora* | - |
| | *P. fortunei* | y = 0.0635x − 68.99 |
| Flowering end | *C. mollissima* | y = 0.7537x − 1358.12 |
| | *M. azedarach* | y = −0.4828x + 1058.91 |
| | *M. grandiflora* | - |
| | *P. fortunei* | y = 0.3064x − 502.499 |

Figure 6 displays the interannual variations in the flowering start and flowering end of woody plants at Guilin. Based on Figure 6 (the upper panel), it can be observed that the flowering start of *C. mollissima*, *M. azedarach*, and *P. fortunei* at the Guilin site did not exhibit a consistent trend. The flowering start of *C. mollissima* and *M. azedarach* remained stable, fluctuating around the 140th day and 100th day of the year, respectively. However, *M. grandiflora* demonstrated a continuous trend of earlier flowering start. On the other

hand, Figure 6 (the bottom panel) reveals that the flowering end of *C. mollissima* exhibited a slight advancement. The flowering end of *M. grandiflora* showed minor variations, mostly fluctuating around the 150th day of the year. *P. fortunei*, except for the period between the early and mid-1980s (1983–1986), exhibited minimal changes with values fluctuating around the 100th day of the year, showing a slight advancement in the long term.

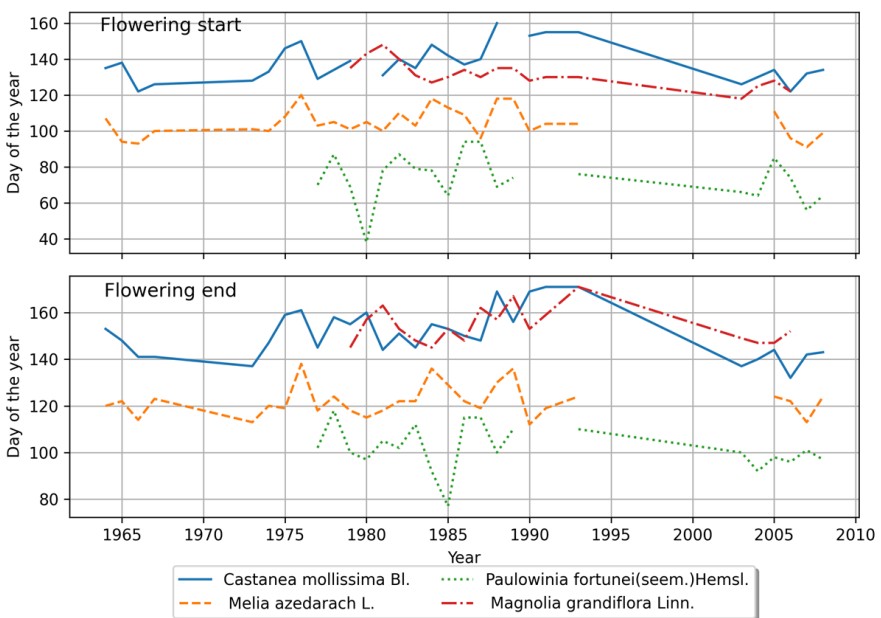

**Figure 6.** The interannual variations of the flowering start and flowering end dates of the woody plants at Guilin.

Table 7 presents the linear regression results of the interannual variation trends of the flowering start and flowering end for woody plant species in Guilin. The average change in flowering start for *C. mollissima* and *M. azedarach* at the Guilin site was less than 1 day per 10 years. However, *P. fortunei* exhibited an average advancement of 2.3 days per 10 years in flowering start, while *M. grandiflora* showed an average advancement of 6.1 days per 10 years. In terms of flowering end, *C. mollissima* and *P. fortunei* exhibited an average advancement of 1.1 and 2.3 days per 10 years, respectively, whereas *M. azedarach* and *M. grandiflora* showed changes of less than 1 day per 10 years. It is evident that the phenological stages of *P. fortunei* at the Guilin site exhibited pronounced variations. Except for *M. grandiflora*, the flowering end of the other three species in Guilin (*C. mollissima*, *M. azedarach*, and *P. fortunei*) showed more changes compared to the flowering start.

**Table 7.** Linear regression analysis of interannual variations in the flowering start and flowering end dates at Guilin.

| Phenological Stages | Species | Linear Regression Equation (Where x Represents Days of the Year) |
|---|---|---|
| Flowering start | *C. mollissima* | y = 0.0114x + 115.064 |
| | *M. azedarach* | y = −0.00634x + 117.06 |
| | *M. grandiflora* | y = −0.6117x + 1347.71 |
| | *P. fortunei* | y = −0.229x + 529.12 |
| Flowering end | *C. mollissima* | y = −0.10897x + 366.56 |
| | *M. azedarach* | y = 0.0439x + 34.96 |
| | *M. grandiflora* | y = −0.0287x + 210.73 |
| | *P. fortunei* | y = −0.2326x + 564.96 |

#### 4.2.3. Comparison of Interannual Variations in Phenological Stages

Tables 8 and 9 present the annual variations in phenological stages in the northern and southern regions of the Nanling Mountains, respectively. The comparison of phenological changes between Changsha and Guilin in Table 8 indicates that both the northern and southern sites showed an overall trend of advancing phenological stages for the studied plants. On the other hand, Table 9 shows that except for a slight advancement in *C. mollissima*'s flowering end at Ganxian, the phenological stages of the other plants demonstrated a trend of delay. Specifically, at Foshan, *C. mollissima* exhibited a delay in both flowering start and flowering end, while *M. azedarach* and *P. fortunei* displayed an advancing trend in their flowering start and flowering end. Comparing the interannual variations in phenological stages among the four plants at Ganxian, Changsha, Foshan, and Guilin, it is evident that except for *C. mollissima* at Foshan, the phenological stages of all plants in the southern region all showed an advancing trend. However, the situation regarding woody plant phenological stages in the northern region was more complex. Specifically, at Changsha in the west, all plants exhibited an advancing trend in the two phenological stages, while at Ganxian in the east, apart from the advancement in *C. mollissima*'s flowering end, the flowering start and flowering end of other plants showed a delaying trend.

**Table 8.** Comparison of phenological changes between Changsha and Guilin.

| Phenological Stages | Species | Changsha | | Guilin | |
| --- | --- | --- | --- | --- | --- |
| | | Changes (d/a) | Trend | Changes (d/a) | Trend |
| Flowering start | *C. mollissima* | −0.0626 | Advancing | −0.188 | Advancing |
| | *M. azedarach* | −0.412 | Advancing | −0.247 | Advancing |
| | *P. fortunei* | - | No trend | −0.339 | Advancing |
| | *M. grandiflora* | −0.776 | Advancing | −0.612 | Advancing |
| Flowering end | *C. mollissima* | −0.902 | Advancing | −0.390 | Advancing |
| | *M. azedarach* | −0.463 | Advancing | −0.01 | Advancing |
| | *P. fortunei* | - | No trend | −0.186 | Advancing |
| | *M. grandiflora* | −0.677 | Advancing | −0.029 | Advancing |

**Table 9.** Comparison of phenological changes between Ganxian and Foshan.

| Phenological Stages | Species | Ganxian | | Foshan | |
| --- | --- | --- | --- | --- | --- |
| | | Changes (d/a) | Trend | Changes (d/a) | Trend |
| Flowering start | *C. mollissima* | 0.377 | Delaying | 1.387 | Delaying |
| | *M. azedarach* | 0.253 | Delaying | −0.272 | Advancing |
| | *P. fortunei* | 0.212 | Delaying | −0.392 | Advancing |
| | *M. grandiflora* | 0.210 | No trend | - | No trend |
| Flowering end | *C. mollissima* | −0.067 | Advancing | 0.261 | Delaying |
| | *M. azedarach* | 0.242 | Delaying | −2.029 | Advancing |
| | *P. fortunei* | 0.671 | Delaying | −0.105 | Advancing |
| | *M. grandiflora* | 0.695 | Delaying | - | No trend |

#### 4.3. Phenological Changes and Climatic Factors

4.3.1. Flowering Start and Climatic Factors

A.  In the northern region

Table 10 displays correlation coefficients between the flowering start dates of woody plants at Ganxian and the key climatic factors, namely monthly or seasonally average temperature, cumulative precipitation, and sunshine duration. The results revealed a significant negative correlation between the flowering start of *C. mollissima* and *M. azedarach* and the average spring temperature, indicating that higher temperatures are associated with earlier phenological stages. Moreover, the monthly average temperature in Febru-

ary demonstrates a significant correlation with the flowering start of *C. mollissima* and *M. azedarach* with that of *C. mollissima* showing a positive correlation and *M. azedarach* showing a negative correlation. In contrast, no significant correlation was found between the phenological stage of *M. grandiflora* and *P. fortunei* and the temperature. April precipitation and February sunshine duration show significant negative correlations for *M. grandiflora* and *M. azedarach*, respectively.

**Table 10.** Correlation coefficients between the flowering start at Ganxian and climatic factors. Significance levels: '*' indicates $p < 0.05$; '**' indicates $p < 0.01$.

| Factors | Species | February | March | April | May | March–April | April–May | Spring (March–May) |
|---|---|---|---|---|---|---|---|---|
| Average temperature | *C. mollissima* | 0.694 ** | −0.356 | −0.451 | 0.185 | −0.560 * | −0.234 | −0.508 * |
| | *M. azedarach* | −0.703 ** | −0.679 | −0.196 | 0.251 | −0.675 ** | −0.009 | −0.599 ** |
| | *M. grandiflora* | 0.274 | −0.068 | 0.133 | 0.253 | 0.024 | 0.212 | 0.11 |
| | *P. fortunei* | −0.345 | −0.031 | −0.025 | −0.092 | −0.039 | 0.033 | 0.001 |
| Precipitation | *C. mollissima* | 0.082 | −0.224 | −0.202 | −0.353 | −0.073 | −0.096 | −0.222 |
| | *M. azedarach* | 0.284 | 0.069 | 0.083 | −0.083 | 0.113 | 0.001 | 0.044 |
| | *M. grandiflora* | −0.058 | 0.261 | −0.672 * | −0.455 | 0.127 | −0.649 * | −0.161 |
| | *P. fortunei* | −0.039 | 0.67 | 0.181 | 0.043 | 0.18 | 0.158 | 0.158 |
| Sunshine duration | *C. mollissima* | −0.393 | 0.271 | −0.221 | 0.213 | 0.085 | 0.056 | 0.235 |
| | *M. azedarach* | −0.479 * | −0.163 | 0.06 | 0.046 | −0.106 | 0.086 | −0.042 |
| | *M. grandiflora* | 0.058 | 0.148 | 0.137 | 0.135 | 0.305 | 0.192 | 0.316 |
| | *P. fortunei* | −0.261 | 0.229 | 0.029 | 0.011 | 0.22 | 0.031 | 0.202 |

Overall, there was a moderately strong correlation ($r = -0.599$, $p < 0.01$) between the flowering start of *M. azedarach* and the average spring monthly temperature, indicating a significant relationship. Figure 7 represents the interannual variation of the flowering start of *M. azedarach* in response to spring temperature. The figure reveals substantial fluctuations in the phenological stages of *M. azedarach* during the period of 1985–1988. Concurrently, the spring temperature also exhibited noticeable variability during these four years.

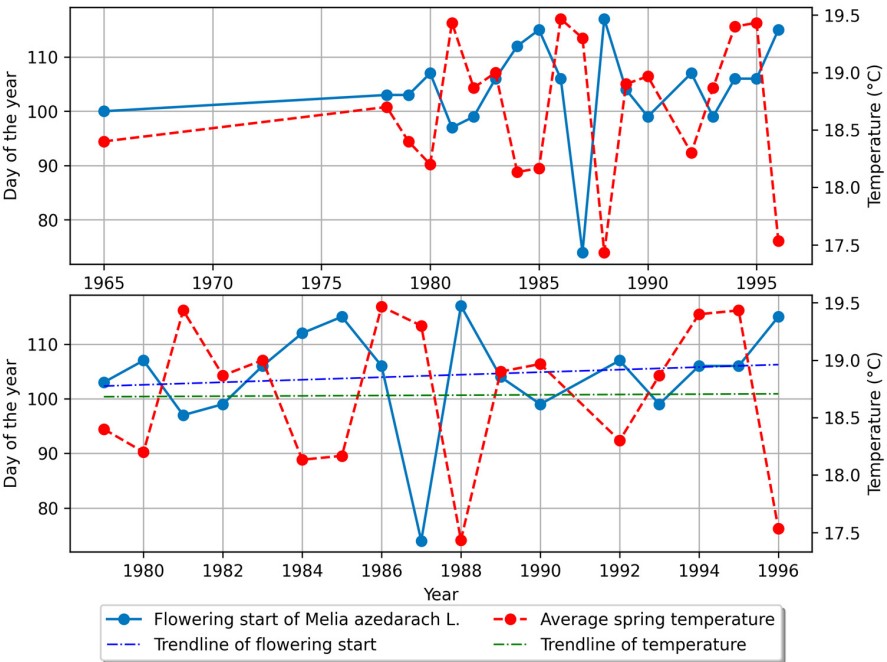

**Figure 7.** Interannual variations of the flowering start of *M. azedarach* and the average spring temperature at Ganxian.

Table 11 presents the correlation between the flowering start of the four studied woody plants at Changsha and the average temperature, sunshine duration, and precipitation. The results showed that *M. grandiflora* and *M. azedarach* exhibit highly significant negative correlations with the average temperature in spring with correlation coefficients of −0.678 ($p < 0.01$) and −0.918 ($p < 0.01$), respectively. This indicates that higher average temperatures are associated with earlier phenological stages. On the other hand, *P. fortunei* and *C. mollissima* show no significant correlation with the average temperature. *P. fortune* exhibits significant correlations with the precipitation in February and *M. azedarach* exhibits significant positive correlations with the precipitation in March. Sunshine duration in spring is highly significant negatively correlated with *M. grandiflora and* significant negatively correlated with *M. azedarach*.

**Table 11.** Correlation coefficients between the flowering start and climatic factors at Changsha. Significance levels: '*' indicates $p < 0.05$; '**' indicates $p < 0.01$.

| Factors | Species | February | March | April | May | March–April | April–May | May–June | Spring (March–May) |
|---|---|---|---|---|---|---|---|---|---|
| Average temperature | *C. mollissima* | −0.167 | −0.373 | −0.495 | −0.38 | −0.499 | −0.510 | −0.366 | −0.503 |
| | *M. azedarach* | −0.237 | −0.551 | −0.726 ** | −0.331 | −0.738 ** | −0.646 ** | - | −0.678 ** |
| | *M. grandiflora* | −0.223 | −0.869 ** | −0.648 * | −0.697 * | −0.908 ** | −0.765 ** | - | −0.918 ** |
| | *P. fortunei* | −0.479 | −0.994 | 0.817 | - | −0.372 | - | - | - |
| Precipitation | *C. mollissima* | −0.125 | 0.511 | −0.475 | −0.022 | −0.70 | −0.325 | 0.029 | −0.066 |
| | *M. azedarach* | −0.218 | 0.602 * | −0.219 | −0.148 | 0.218 | −0.245 | - | 0.038 |
| | *M. grandiflora* | −0.085 | 0.493 | 0.067 | −0.015 | 0.408 | 0.035 | - | 0.300 |
| | *P. fortunei* | 0.985 * | −0.741 | 0.027 | - | −0.329 | - | - | - |
| Sunshine duration | *C. mollissima* | −0.409 | −0.341 | −0.195 | −0.238 | −0.373 | −0.329 | −0.091 | −0.414 |
| | *M. azedarach* | −0.153 | −0.461 | −0.556 * | −0.164 | −0.686 ** | −0.513 * | - | −0.601 * |
| | *M. grandiflora* | −0.077 | −0.779 ** | −0.156 | −0.540 | −0.603 * | −0.603 * | - | −0.801 ** |
| | *P. fortunei* | −0.886 | −0.683 | 0.931 | - | −0.024 | - | - | - |

To facilitate comparison and analysis with the Ganxian station, the interannual variation curve was plotted for *M. azedarach*'s flowering start and the spring average temperature. Figure 8 illustrates a negative overall correlation between *M. azedarach*'s flowering start and the spring average temperature, although it was exhibiting more complex patterns during different time periods. *M. azedarach*'s flowering start showed an increasing trend during the period of 1983–1991. The interannual variation in the spring average temperature revealed a decreasing trend at Changsha during the same period. In close comparison to the observations at Ganxian, these alterations correspond to a decrease in temperature, resulting in a delayed flowering start for the *P. fortunei*. However, in the period of 2003–2008, despite a sudden rise in the spring average temperature, there was an exacerbation of the delay in *M. azedarach*'s flowering start. This finding shows that at Changsha, *M. azedarach*'s flowering start displayed a delaying trend during a gradual decline in average temperature, but interestingly, it also exhibited a delaying trend in flowering despite a sharp increase in average temperature, emphasizing the need to consider the effects of other climatic factors and the plant's own adaptation strategies.

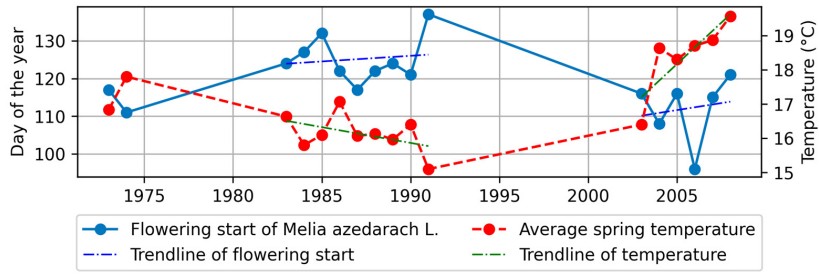

**Figure 8.** Interannual variations of the flowering start of *M. azedarach* and the average spring temperature at Changsha.

B. In the southern region

Correlation between the flowering start and climate factors at Foshan, which is in the southern region of the Nanling Mountains, is presented in Table 12. There was a significant negative correlation between the flowering start of *M. azedarach* and the temperature in February (−0.535, $p < 0.01$) as well as in March (−0.554, $p < 0.01$). On the other hand, the flowering start of *P. fortunei*, *C. mollissima*, and *M. grandiflora* showed weak correlations with the average temperature, and the influence of sunshine duration and precipitation on the phenological stage of these four woody plant species was also very weak.

**Table 12.** Correlation coefficients between the flowering start and climatic factors at Foshan. Significance levels: '*' indicates $p < 0.05$; '**' indicates $p < 0.01$.

| Factors | Species | February | March | April | May | Feburary–March | March–April | April–May | Spring (March–May) |
|---|---|---|---|---|---|---|---|---|---|
| Average temperature | *C. mollissima* | −0.278 | −0.361 | −0.218 | 0.495 | −0.384 | −0.367 | 0.157 | −0.141 |
| | *M. azedarach* | −0.535 ** | −0.301 | - | - | −0.554 ** | - | - | - |
| | *M. grandiflora* | −0.539 | 0.324 | −0.665 | - | −0.445 | −0.269 | - | - |
| | *P. fortunei* | −0.379 | −0.492 | - | - | −0.537 | - | - | - |
| Precipitation | *C. mollissima* | 0.617 | 0.141 | −0.446 | −0.431 | 0.539 | −0.319 | −0.537 * | −0.452 |
| | *M. azedarach* | 0.213 | 0.144 | - | - | 0.207 | - | - | - |
| | *M. grandiflora* | 0.169 | 0.022 | 0.556 | - | 0.12 | 0.445 | - | - |
| | *P. fortunei* | 0.213 | 0.144 | - | - | 0.207 | - | - | - |
| Sunshine duration | *C. mollissima* | −0.197 | 0.186 | −0.177 | 0.371 | 0.004 | 0.034 | 0.163 | 0.183 |
| | *M. azedarach* | 0.033 | 0.265 | - | - | 0.184 | - | - | - |
| | *M. grandiflora* | 0.033 | 0.265 | −0.716 | - | 0.184 | −0.47 | - | - |
| | *P. fortunei* | −0.031 | 0.19 | - | - | 0.079 | - | - | - |

Figure 9 displays the yearly variations in the flowering start of *M. azedarach* in Foshan along with the corresponding average temperature during spring. From 1976 to 1990, the average spring temperature in Foshan remained stable, and the flowering start of *M. azedarach* showed no noticeable changes. However, between 2003 and 2008, Foshan experienced a decrease in average temperature, resulting in a noticeable delay in the flowering start. Similar trends were observed above in the northern region of the Nanling Mountains. These findings shows that the flowering start of *M. azedarach* is primarily influenced by the average temperature, and this relationship holds true for both the northern and southern regions of the Nanling Mountains.

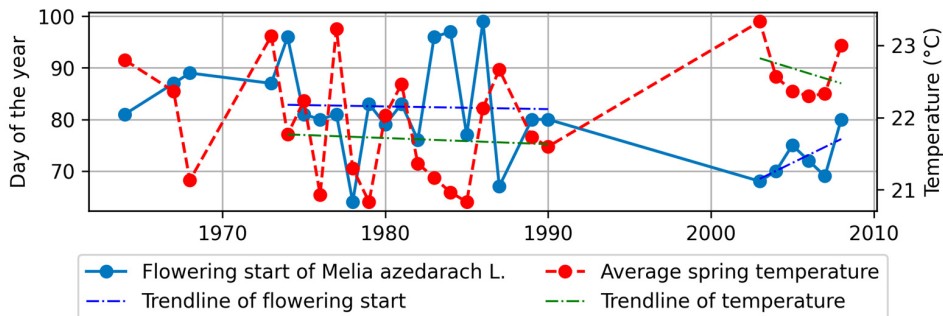

**Figure 9.** Interannual variations of the flowering start of *M. azedarach* and the average spring temperature at Foshan.

The relation analysis at Guilin is summarized in Table 13. The flowering start of *C. mollissima* and *M. azedarach* showed the strongest correlations with the spring average temperature with correlation coefficients of −0.734 ($p < 0.01$) and −0.551 ($p < 0.01$), respectively, indicating a highly significant and negative association. Specifically, the flowering start of *M. azedarach* was notably influenced by the monthly average temperature of the preceding month, i.e., February, while the flowering start of *C. mollissima* was highly influ-

enced by the average temperature of March and April. Furthermore, the flowering timing of *M. grandiflora* was notably affected by the average temperature in April. *M. grandiflora* exhibits highly significant positive correlations with the precipitation in April. Sunshine duration in March is significant negatively correlated with *M. grandiflora, M. azedarach* and *C. mollissima.*

**Table 13.** Correlation coefficients between the flowering start and climatic factors at Guilin. Significance levels: '*' indicates $p < 0.05$; '**' indicates $p < 0.01$.

| Factors | Species | February | March | April | May | March–April | April–May | May–June | Spring (March–May) |
|---|---|---|---|---|---|---|---|---|---|
| Average temperature | *C. mollissima* | −0.138 | −0.529 ** | −0.489 ** | −0.376 * | −0.635 ** | −0.589 ** | −0.150 | −0.734 ** |
| | *M. azedarach* | −0.514 ** | −0.621 ** | −0.218 | - | −0.548 ** | - | - | −0.551 ** |
| | *M. grandiflora* | −0.161 | −0.207 | −0.524 * | −0.240 | −0.413 | −0.475 * | - | −0.471 * |
| | *P. fortunei* | −0.097 | 0.120 | 0.185 | - | 0.173 | - | - | 0.098 |
| Precipitation | *C. mollissima* | 0.341 | 0.032 | −0.211 | 0.059 | −0.153 | −0.067 | −0.010 | −0.049 |
| | *M. azedarach* | −0.029 | −0.199 | −0.198 | - | −0.257 | - | - | −0.024 |
| | *M. grandiflora* | 0.189 | 0.169 | 0.605 ** | −0.329 | 0.564 * | 0.170 | - | 0.213 |
| | *P. fortunei* | −0.190 | 0.210 | 0.074 | - | 0.146 | - | - | 0.291 |
| Sunshine duration | *C. mollissima* | −0.298 | −0.465 * | −0.118 | −0.342 | −0.384 * | −0.307 | −0.089 | −0.437 * |
| | *M. azedarach* | −0.370 | −0.408 * | −0.097 | - | −0.335 | - | - | −0.372 |
| | *M. grandiflora* | −0.192 | −0.504 * | −0.343 | 0.054 | −0.587 ** | −0.220 | - | −0.435 |
| | *P. fortunei* | −0.088 | 0.179 | −0.046 | - | 0.084 | - | - | 0.119 |

Figure 10 presents the interannual variation curves of Guilin's flowering start of *M. azedarach* and the average spring temperature. The results are consistent with the findings presented in Figures 7–9. Specifically, during the period from 1970 to 1990, there was a noticeable lagged trend (upward) in the flowering start of *M. azedarach* when the average spring temperature decreased.

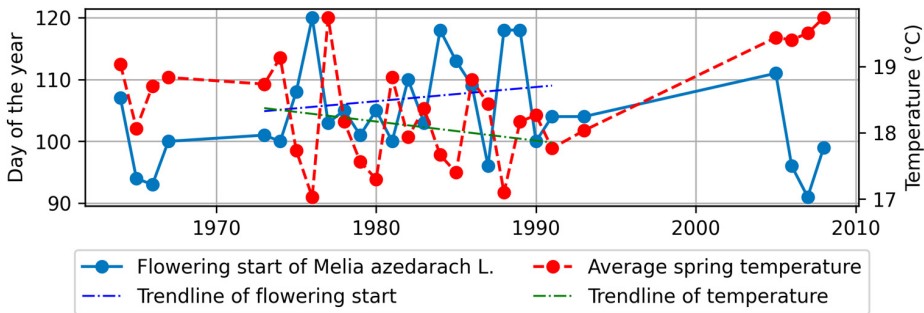

**Figure 10.** Interannual variations of the flowering start of *M. azedarach* and the average spring temperature at Guilin.

4.3.2. Flowering End and Climate Factors

A. In the northern region

The correlation analysis between the flowering end of the four studied woody plants (*C. mollissima, P. fortunei, M. azedarach,* and *M. grandiflora*) at Ganxian in the northern region of the Nanling Mountains and climatic factors indicated significant relationships (Table 14). The flowering end of *M. azedarach* exhibited the strongest correlation with spring average temperature (correlation coefficient: −0.719, $p < 0.01$), demonstrating a clear negative association. The flowering end of *C. mollissima* also exhibited a significant positive correlation with spring average temperature with correlation coefficients of 0.512 ($p < 0.05$). However, no significant correlations were found between the flowering end of *P. fortunei*, *M. grandiflora*, and average temperature. Moreover, the influence of sunshine duration and precipitation on the flowering end of the four woody plants was weak.

**Table 14.** Correlation coefficients between the flowering end at Ganxian and climatic factors. Significance levels: '*' indicates $p < 0.05$; '**' indicates $p < 0.01$.

| Factors | Species | March | April | May | June | March–April | April–May | May–June | Spring (March–May) |
|---|---|---|---|---|---|---|---|---|---|
| Average temperature | *C. mollissima* | −0.312 | −0.334 | −0.067 | 0.013 | −0.455 | −0.284 | −0.027 | 0.512 * |
| | *M. azedarach* | −0.558 * | −0.486 * | 0.158 | - | −0.748 ** | −0.275 | - | −0.719 ** |
| | *M. grandiflora* | 0.025 | 0.087 | −0.453 | −0.013 | 0.094 | −0.132 | −0.237 | −0.106 |
| | *P. fortunei* | −0.005 | −0.027 | - | - | 0.012 | 0.015 | - | 0.008 |
| Precipitation | *C. mollissima* | −0.001 | −0.25 | −0.244 | −0.016 | −0.166 | −0.33 | −0.144 | −0.239 |
| | *M. azedarach* | 0.006 | −0.218 | −0.259 | - | −0.139 | −0.319 | - | −0.227 |
| | *M. grandiflora* | 0.237 | −0.431 | −0.086 | 0.409 | 0.136 | −0.222 | 0.268 | 0.055 |
| | *P. fortunei* | −0.048 | −0.025 | - | - | −0.058 | −0.167 | - | −0.152 |
| Sunshine duration | *C. mollissima* | 0.004 | 0.015 | −0.178 | −0.6 | 0.015 | −0.162 | −0.145 | −0.132 |
| | *M. azedarach* | −0.066 | −0.028 | −0.125 | - | −0.083 | −0.14 | - | 0.163 |
| | *M. grandiflora* | 0.569 | −0.519 | −0.412 | −0.323 | 0.220 | −0.625 | −0.458 | −0.225 |
| | *P. fortunei* | 0.464 | −0.171 | - | - | 0.274 | −0.142 | - | 0.221 |

At Changsha in the western region (Table 15), the influence of climatic factors on the flowering end was found to be intricate. Notably, the flowering end of *M. azedarach* exhibited a remarkably strong negative correlation coefficient of −0.686 ($p < 0.01$) with the spring average temperature, indicating a significant inverse relationship. Furthermore, other climatic factors, including sunshine duration and precipitation, also impacted the flowering end of plants. Specifically, there was a significant negative correlation coefficient of −0.695 ($p < 0.01$) between precipitation in May–June and *C. mollissima*, a correlation coefficient of −0.641 ($p < 0.01$) between sunshine duration in April and *M. azedarach*'s flowering end, and also a correlation coefficient of −0.769 ($p < 0.01$) between sunshine duration in May–June and *M. grandiflora*'s flowering end, all indicating significant negative associations. In comparison, *M. azedarach*'s flowering end demonstrated a highly significant symmetry with the spring average temperature and sunshine duration in April. Additionally, *C. mollissima*'s flowering end showed a significant negative correlation with precipitation in May–June, suggesting that higher May–June precipitation led to a later flowering end of *C. mollissima*.

**Table 15.** Correlation coefficients between the flowering end and climatic factors at Changsha. Significance levels: '*' indicates $p < 0.05$; '**' indicates $p < 0.01$.

| Factors | Species | March | April | May | June | March–April | April–May | May–June | Spring (March–May) |
|---|---|---|---|---|---|---|---|---|---|
| Average temperature | *C. mollissima* | −0.079 | −0.572 * | −0.119 | −0.563 * | −0.366 | −0.428 | −0.345 | −0.316 |
| | *M. azedarach* | −0.550 * | −0.734 ** | −0.345 | - | −0.742 ** | −0.658 ** | - | −0.686 ** |
| | *M. grandiflora* | −0.691 | −0.304 | −0.403 | −0.685 * | −0.619 | −0.391 | −0.597 | −0.586 |
| | *P. fortunei* | −0.922 | 0.237 | - | - | −0.893 | - | - | - |
| Precipitation | *C. mollissima* | 0.194 | 0.262 | −0.691 ** | −0.356 | 0.374 | −0.381 | −0.695 ** | −0.275 |
| | *M. azedarach* | 0.563 * | −0.209 | −0.212 | - | 0.2 | −0.286 | - | −0.02 |
| | *M. grandiflora* | 0.352 | 0.230 | 0.056 | −0.051 | 0.472 | 0.210 | −0.002 | 0.396 |
| | *P. fortunei* | −0.871 | 0.073 | - | - | −0.338 | - | - | - |
| Sunshine duration | *C. mollissima* | −0.026 | −0.399 | 0.369 | 0.058 | −0.34 | 0.034 | 0.382 | 0.017 |
| | *M. azedarach* | −0.45 | −0.504 * | −0.132 | - | −0.641 ** | −0.451 | - | −0.547 * |
| | *M. grandiflora* | −0.509 | −0.298 | −0.387 | −0.405 | −0.538 | −0.47 | −0.769 ** | −0.556 |
| | *P. fortunei* | −0.980 * | 0.553 | - | - | −0.680 | - | - | - |

**B.　In the southern region**

According to the correlation analysis between the flowering ends at Foshan in the southern region of Nanling Mountains and climatic factors (Table 16), the strongest correlation was observed between the flowering ends of *P. fortunei* and the March–April average temperature, showing a highly significant negative correlation coefficient of −0.808 ($p < 0.01$). The flowering end of *C. mollissima* was highly influenced by the sunshine dura-

tion in April ($-0.592$). However, the sunshine duration and precipitation in March–June had minor impacts on the flowering end of the four woody plants.

**Table 16.** Correlation coefficients between the flowering end and climatic factors at Foshan. Significance levels: '*' indicates $p < 0.05$; '**' indicates $p < 0.01$.

| Factors | Species | March | April | May | March–April | April–May | May–June | Spring (March–May) |
|---|---|---|---|---|---|---|---|---|
| Average temperature | *C. mollissima* | 0.011 | −0.162 | 0.128 | −0.075 | −0.028 | - | −0.013 |
| | *M. azedarach* | −0.02 | −0.173 | −0.061 | −0.11 | −0.159 | - | −0.125 |
| | *M. grandiflora* | 0.493 | 0.465 | −0.697 | 0.71 | −0.125 | - | 0.274 |
| | *P. fortunei* | −0.737 ** | −0.593 * | - | −0.808 ** | - | - | - |
| Precipitation | *C. mollissima* | −0.288 | −0.197 | −0.548 | −0.293 | −0.536 | - | −0.523 |
| | *M. azedarach* | −0.045 | 0.12 | −0.207 | 0.068 | −0.133 | - | −0.138 |
| | *M. grandiflora* | −0.478 | −0.241 | 0.034 | −0.398 | −0.039 | - | −0.104 |
| | *P. fortunei* | −0.478 | −0.241 | - | −0.398 | - | - | - |
| Sunshine duration | *C. mollissima* | 0.235 | −0.592 * | 0.108 | −0.105 | −0.187 | - | −0.012 |
| | *M. azedarach* | 0.204 | −0.166 | −0.001 | 0.031 | −0.085 | - | 0.019 |
| | *M. grandiflora* | 0.647 | −0.012 | −0.08 | 0.716 | −0.057 | - | 0.325 |
| | *P. fortunei* | −0.279 | −0.069 | - | −0.178 | - | - | - |

The correlation analysis from the Guilin station in the western region revealed important associations between the phenology of selected woody plants and climatic factors (Table 17). Specifically, the flowering end of *C. mollissima* exhibited a strong negative correlation with the average temperature in spring (correlation coefficient = $-0.713$, $p < 0.01$). Similarly, the flowering end of *M. azedarach* showed a significant negative correlation with the average temperature in March (correlation coefficient = $-0.485$, $p < 0.01$). Additionally, precipitation in March had a noticeable impact on the flowering end of *M. azedarach* (correlation coefficient = $-0.390$). Furthermore, the flowering end of *M. grandiflora*, occurring in May–June, demonstrated a significant negative correlation with the sunshine duration in the preceding months of March and April (correlation coefficient = $-0.587$, $p < 0.01$).

**Table 17.** Correlation coefficients between the flowering end and climatic factors at Guilin. Significance levels: '*' indicates $p < 0.05$; '**' indicates $p < 0.01$.

| Factors | Species | March | April | May | June | March–April | April–May | May–June | Spring (March–May) |
|---|---|---|---|---|---|---|---|---|---|
| Average temperature | *C. mollissima* | −0.462 * | −0.508 ** | −0.415 * | 0.188 | −0.585 ** | −0.639 ** | −0.219 | −0.713 ** |
| | *M. azedarach* | −0.485 ** | −0.224 | 0.043 | - | −0.465 * | −0.137 | - | −0.399 * |
| | *M. grandiflora* | 0.172 | −0.246 | −0.204 | −0.192 | −0.001 | −0.273 | −0.262 | −0.079 |
| | *P. fortunei* | 0.286 | −0.084 | - | - | 0.166 | - | - | - |
| Precipitation | *C. mollissima* | 0.121 | −0.024 | 0.119 | −0.110 | 0.039 | 0.082 | 0.012 | 0.106 |
| | *M. azedarach* | −0.390 * | −0.208 | 0.115 | - | −0.360 | −0.019 | - | −0.129 |
| | *M. grandiflora* | 0.168 | 0.430 | −0.204 | 0.247 | 0.429 | 0.145 | 0.067 | 0.192 |
| | *P. fortunei* | 0.285 | 0.246 | - | - | 0.312 | - | - | - |
| Sunshine duration | *C. mollissima* | −0.444 * | −0.156 | −0.237 | 0.210 | −0.391 * | −0.253 | −0.028 | −0.387 * |
| | *M. azedarach* | −0.211 | 0.002 | −0.046 | - | −0.137 | −0.032 | - | −0.112 |
| | *M. grandiflora* | −0.504 * | −0.343 | 0.054 | 0.252 | −0.587 ** | −0.220 | 0.241 | −0.435 |
| | *P. fortunei* | 0.188 | −0.224 | - | - | −0.056 | - | - | - |

## 5. Discussion

Based on the above analysis, four key findings were observed. Firstly, different plant species in the same region showed varying sensitivity to climatic factors. Specifically, the flowering start of *C. mollissima* and *M. azedarach* was highly influenced by spring average temperature. *M. azedarach* exhibited the highest sensitivity to climatic changes in terms of phenological stages. Secondly, within the same plant species, different phenological stages responded differently to climatic factors. For example, the flowering start of *C. mollissima*

was strongly influenced by spring average temperature, while its flowering end did not show a clear correlation with this factor. Similarly, the flowering start of *M. azedarach* displayed a significant negative correlation with the average temperature in early March, but its flowering end did not show a clear relationship with the average temperature during the same period. Thirdly, different climatic factors had varying effects on different plants. Among these factors, average temperature was identified as the key driver of phenological changes in response to climate variations. The phenological stages of plants showed a significant negative correlation with spring temperatures, indicating an advancing trend in phenological stages with higher temperatures. Lastly, even within the same plant species, the influence of climatic factors on phenological stages varied across different regions. Specifically, the flowering start of *C. mollissima* showed a significant correlation with spring average temperature in Ganxian and Guilin stations, while no significant correlation was found in the Changsha and Foshan stations.

It was observed that the phenological stages of woody plants in the southern region of the Nanling Mountains occurred 2–31 days earlier than those in the northern region. Moreover, an advancing trend of 0.1 to 2 days per decade in phenological stages was observed in the southern region of the Nanling Mountains. The same case was found by Ding et al. (2022), whose research revealed that the growing season (SOS) in the southern part of the Nanling Mountains occurred slightly earlier than in the central and northern regions [27]. Similar results have also been documented in Europe, where Menzel et al. (2006) analyzed phenological observation data from various European locations, including flowering, leaf emergence, and fruit ripening times, and found an advancing trend of 2.5 days per decade in phenological stages across different regions in response to warming climates [28]. These findings suggest that plant phenology in the study area is evidently influenced by climate changes and keeps the similar rate with other parts of the earth. Furthermore, the impacts of climate change on natural systems exhibit global consistency. Parmesan and Yohe (2003) collected phenological data from multiple research sites globally and compared them with climate data [15]. They identified similar trends (2.3 days per decade) and patterns in species responses to climate change across different regions and ecosystems, such as advancing phenological stages, altered animal migration patterns, and accelerated glacier melting. This study serves as another example of the influence of climate change on global natural systems.

The observed variations in plant phenology along the north–south gradient of the Nanling Mountains can be attributed not only to climate change but also to other factors, including extreme weather events, elevation, latitude, and longitude within the mountainous environment. We found that in 1985, noteworthy events unfolded in Ganxian involving *P. fortunei*, which was characterized by an earlier flowering start and flowering ends. Moreover, the year 1988 witnessed a notable peak in all four species with two of them reaching their maximum values, reflecting a delay in the start and flowering end. Similar occurrences were observed in Changsha in 2005 (see Figure 4), Foshan from 1977 to 1980 (see Figure 5), and Guilin from 1980 to 1985 (see Figure 6). Wang et al. (2016) reported profound impacts of extreme cold weather in 2008 on butterfly communities in the Nanling Mountains [29]. The chilly weather resulted in a noticeable reduction in temperate and tropical butterfly species, leading to changes in community structure with temperate species dominating the community after disturbances. This effect persisted for two years until the abundance of tropical species recovered to pre-disturbance levels. Wolkovich et al. (2014) conducted long-term observations and analyses of plant phenological stages in the Canadian Rocky Mountains [30]. They evaluated the relationship between these phenological events, such as flowering start and leaf senescence, and environmental factors such as latitude, elevation, and temperature using data collected from various sites at different elevations and latitudes. The results indicated that plants in higher latitude and elevation regions exhibited shorter phenological stages, while those in lower latitude and elevation regions displayed longer phenological stages. Singh et al. (2015) observed that all phenological events of *Rhododendron arboreum* Sm. in the central Himalayas initiate earlier

at lower elevations and are delayed at higher elevations [31]. However, a detailed analysis focusing on the effect of latitude and other factors was constrained in this study due to insufficient information. Future research with more extensive data could provide valuable insights into this aspect.

This study reveals variations in the responses of the two phenological stages (flowering start and end) to climate changes. At Foshan, a significant correlation emerged between the flowering start of *M. azedarach* and February–March average temperature. However, no significant correlation was observed between the flowering end of *M. azedarach* and the average temperature in the same period. Similarly, the flowering end of *P. fortunei* exhibited a significant correlation with the March–April average temperature, whereas the flowering start of this species showed no clear correlation with the average temperature. At Changsha, the flowering end of *C. mollissima* showed a significant correlation with precipitation, while the flowering start showed no notable correlation with precipitation. Therefore, it can be hypothesized that the changes in phenological stages are primarily influenced by a specific sensitive climatic factor during a given period. For instance, the fluctuations in the flowering start of *M. azedarach* at Ganxian can be attributed to variations in spring temperature: a decrease in temperature leads to a delay in the flowering start, while an increase in temperature results in an advancement. These findings align with the research conducted by Fu et al. (2015) who found that spring phenological events, such as flowering start, were more sensitive to temperature changes, while summer and autumn phenological events, such as flowering end and leaf senescence, were more sensitive to precipitation changes [10]. Moreover, Piao et al. (2015) conducted a global study on the relationship between plant phenological stages and temperature changes, and they also found variations in the responses of varied species to climate change. Early spring flowering plants exhibited greater sensitivity to temperature increase, while late spring and summer flowering plants showed weaker responses to temperature changes [32]. Latitude position also influences the response of plant phenological stages to climate change. Badeck et al. (2004) suggested that in mid-to-high latitude regions, temperature serves as a primary driving factor for many plant developmental processes, and higher temperatures tend to accelerate plant development, resulting in an early transition to the next developmental stage [33].

This study provides crucial evidence for understanding the mechanisms underlying the response of plant phenological stages to climate change. By examining the relationship between phenological stages and climatic factors in varied species across the north and south sides of the same region, we have revealed the diversity in the response of the two plant phenological stages (flowering start and end) to climate change, which enhances our understanding of phenological ecology and plant adaptability. The differential sensitivity of different phenological stages to climate change and the varying responses of varied species to climatic factors contribute to advancing our knowledge in the fields of phenology ecology and climate change. Additionally, this study provides valuable insights for plant phenological prediction and adaptive management in different geographical regions.

## 6. Conclusions

This study utilized the phenological observation data of four plants, namely *C. mollissima*, *P. fortunei*, *M. azedarach*, and *M. grandiflora*, collected from four stations (Guilin, Changsha, Ganxian, and Foshan) on both sides of the Nanling Mountains, covering different time periods from 1963 to 2008. Additionally, climate data including monthly average temperature, monthly precipitation, and sunshine duration from the nearest meteorological stations were incorporated. The research employed methods such as linear trend estimation, Pearson correlation analysis, and comparative analysis to investigate the trends in plant phenology and their relationships with changing climatic factors. The main conclusions are as follows:

(1) During the study period, the studied phenological stages of woody plants in the southern region of the Nanling Mountains occurred 2–31 days earlier than those in the northern region. Moreover, a noticeable advancing trend of 0.1 to 2 days per decade

was observed in the phenological stages of all plants in the southern region of the Nanling Mountains.

(2) Among the four stations, *M. azedarach* exhibited the highest sensitivity to climatic factor changes, specifically in terms of flowering start and flowering end. Notably, there was a significant negative correlation between the phenological stages of *M. azedarach* and spring temperatures, indicating that higher temperatures led to earlier phenological stages.

(3) The influence of the same climatic factor on different phenological stages of the same plant species varied. For example, at Foshan, the flowering start of *M. azedarach* exhibited a significant correlation with the average temperature of February–March, whereas the flowering end of the same species did not show a clear correlation with average temperature. Similarly, the flowering end of *P. fortunei* exhibited a significant correlation with the average temperature of March–April, while the correlation with average temperature was less evident for the flowering start of the same species. Moreover, the flowering end of *C. mollissima* at Changsha exhibited a significant correlation with precipitation, whereas the correlation with precipitation was not significant for the flowering start.

(4) Different regions displayed varying degrees of influence from climatic factors on plant phenology. For instance, the phenological stages of woody plants at Foshan and Ganxian were affected by the average temperature in May, while other climatic factors had less impact. On the other hand, the phenological stages of the plants at Changsha and Guilin were influenced by sunshine duration and precipitation. Specifically, the flowering end of *C. mollissima* at Changsha displayed a significant negative correlation with precipitation in May–June, the flowering start of *M. grandiflora* at Changsha exhibited a significant correlation with spring sunshine duration, and both phenological stages of *M. grandiflora* at Guilin showed a significant negative correlation with sunshine duration in March–April.

**Author Contributions:** G.L. conceived and designed the study. G.L., A.X., H.L. (Haihui Lv) and Z.H. collected and analyzed the data. G.L., H.L. (Haihui Lv) and Z.W. interpreted the results. H.L. (Haolong Liu) and L.C. helped with data collection and analysis. G.L. and H.L. (Haolong Liu) contributed to the manuscript preparation. All authors have read and agreed to the published version of the manuscript.

**Funding:** This research was funded by the Humanities and Social Science Research Planning Project for Universities of Jiangxi Province, grant number GL20116; Science and Technology Project of Jiangxi Department of Education, grant number GJJ201419; and National Natural Science Foundation of China, grant number 42361011.

**Data Availability Statement:** The underlying research data for this study are available upon request from the corresponding author (Guangxu Liu, liuguangxu@gnnu.edu.cn). In the interest of transparency and reproducibility, we are committed to sharing our data with interested researchers. To request access to the data, readers can contact the corresponding author and provide a brief description of their intended use. We will review all requests and aim to respond within a reasonable time limit. Alternatively, readers can access a subset of the data used in this study through the National Earth System Science Data Center, National Science & Technology Infrastructure of China at http://www.geodata.cn/data/datadetails.html?dataguid=6727090&docid=26123 (accessed on 30 March 2023) and the National Meteorological Information Center–China Meteorological Data Network at http://data.cma.cn/data/detail/dataCode/A.0012.0001.html (accessed on 30 March 2023). We encourage readers to use the data to verify our findings and to build on our research.

**Acknowledgments:** Acknowledgement also for the data support from National Earth System Science Data Sharing Infrastructure, National Science & Technology Infrastructure of China (http://www.geodata.cn, accessed on 29 January 2021).

**Conflicts of Interest:** We have declared that no competing interests exist and consent to publication of this manuscript.

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
