# Peer review of "Phenological Changes of Woody Plants in the Southern and Northern Regions of Nanling Mountains and Their Relationship with Climatic Factors"

_forests, doi:10.3390/f14122363_

Round 1

Reviewer 1 Report

Comments and Suggestions for Authors

Abstract

- line 17 - add sentence ... spanning from 1963 to 2008 ...in different time periods

- line 21 ... detto, since the time series is not continuous

- line 22 - add numerical data on the shift of trends in the northern and southern regions 

- revise the abstract according to the added results

- to add any additional results that emerge after the modification and addition of further results.

The name of the tree Melia azedarach L. is also incorrectly given throughout the thesis as Melia azedarace : (Abstract, Fig.1-5 Tab.1-5, text ... etc.) The correct name needs to be unified.

The thesis deals with the current issue of indigenous tree species in the Nailing Mts in relation to environmental conditions. These are evaluated as factors (average temperature, precipitation, sunshine duration). Only in some parts of the thesis, the long time series are evaluated, which is important for assessing the further development of ecosystems under conditions of climate change.

The title of the thesis partly corresponds to its content, as only short phenological time series are evaluated in several chapters of the results. These do not represent the response of tree species to climate change. I therefore recommend a modification of the title: to have the section ' A study based on long-term observations' removed.

1.    Introduction

The introduction is disjointed and unconnected in thought. Many parts do not address the issue of the impact of climate change on native forest ecosystems and the influence of environmental factors. I recommend omitting inappropriate parts of the chapter:

- line 70:Jochner-Menzel (2015) addresses urban phenology

- line 72: Sun et al. (2016) analyzed water use efficiency models using satellite data and carbon cycle models

- line 89: trends related to agriculture

- line 125: the work of Wang et al. (2016) does not represent climate change, but is useful for discussion - for comparison what was the trend of the assessed environmental factors in 2008

The Introduction needs to be studied further, revised and supplemented with topics addressed in the following works from other parts of the world, e.g. Europe, Sev. America (or others with the given topic). The knowledge that the authors of this article gain by studying the recommended papers can be used to improve the quality of the very brief discussion, e.g. these papers :

Asse, D. et al. (2018). Warmer winters reduce the advance of tree spring phenology induced by warmer springs in the Alps. Agricultural and Forest Meteorology 252, 220230.

Lukasová, V. et al. (2019). Validation and application of European beech phenological metrics derived from MODIS data along an altitudinal gradient. Forests10(1), 60.

Škvareninová, J., Mrekaj, I. (2022). Impact of Climate Change on Norway Spruce Flowering in the Southern Part of the Western Carpathians. Frontiers in Ecology and Evolution
Babálová, D. et al. (2018). The dynamics of the phenological development of four woody species in south-west and central Slovakia. Sustainability

Partanen, J. et al. (1998). Effects of photoperiod and temperature on the timing of bud burst in Norway spruce (Picea abies). Tree Physiology, 18:811–816.

Beaubien, E. G., Freeland, H. J. (2000). Spring phenology trends in Alberta. Canada: links to ocean temperature. Int J Biometeorol. 44(2), 53–59.

Saderi, S. et al.  (2019). Phenology of wood formation in larch (Larix decidua Mill.) trees growing along a 1000-m elevation gradient in the French Southern Alps. Annals of Forest Science76(3):117.

Ahas, R. et al.  (2002). Changes in European spring phenology. Int. J. Climatol. 22, 1727–1738.

Wenden, B. et al. (2020). Shifts in the temperature-sensitive periods for spring phenology in European beech and pedunculate oak clones across latitudes and over recent decades. Global Change Biology 26, 1808–1819.

2. Study area

- The chapter lacks the elevations at which the observed species grow, or the elevations of the observation stations, the approximate ages of the trees in the northern and southern Nanling Mts.

- Fig.1 - add the location of the stations

- Given the analysis of the results, a more detailed analysis of the climate of the mountains is also needed because the results in section 4.1.3 assess the phenological phase in 2 regions (north-south) and in 2 areas (east-west), which may significantly change the results. What are the climate differences (temperature -precipitation) between regions and areas. Please summarize all comments in a clear table and analyze it in the results.

3. Data and Methods

- A precise definition of the phenological phases to be assessed should be added to the chapter: a description of the degree of the phase beginning and end of flowering should be given - e.g. % of flowers or a phenological manual with a scale should be used and quoted.

- Specify the object of the survey - the number of trees of each species observed or other areal indication of the sample observed.

- Indicate the altitude of the stations or trees observed

- Indicate missing years and divide years of observation into short, medium and long time series - summarise in a table by tree species

- Filling in missing data from 1963-2008 average does not give correct results, can only be done for individual missing years and not long time series (10-15 years)

4.Results

4.1.1 Characteristics of the plant phenology in the northern region

A more detailed flowering analysis needs to be done and the results need to be completed:

- by altitude, which affects the onset and duration of phenophases in the northern region. The average values from stations without this factor also give biased results compared to the southern exposure

- provide basic statistical characteristics of the variability for individual species in a given region and altitude

- evaluate only the trends of long time series species (Melia azedarach, Castanea mollissima) and compare them by exposure, altitude and latitude, respectively

- by latitude - if sites are at a significant distance

- to analyse the flowering results in more detail according to the above parameters

4.1.2 same as 4.1.1

4.1.3

Tables 1, 2 provide a comparison of plant phenology in the southern and northern regions of the eastern Nanling Mts. and western Nanling Mts. which leads to confusion. The title of the paper contains a comparison of woody plants in the southern and northern regions of Nanling Mts. The data presented needs to be suitably supplemented and modified. I recommend that all the above results should be reviewed in more detail by region and area.

4.2.

4.2.1

Trends express the course of long-term time series. Therefore, it is necessary to modify the terminology and to use this term only when assessing the progression of a phase over a period of approximately 20 years or more: therefore, trends can only be assessed in Fig. 2 and Fig. 3 for the species Castanea mollissima and Melia azedarach. It would be useful to include the trend and statistical significance in the graph for the species mentioned above.

The phenological data for Paulownia fortunei and Magnolia grandiflora are not suitable for assessment in relation to climate change as the time series are short and discontinuous, filling in long missing data with average values does not give correct results.

- line 377 – incorrect wording, it is not inter-annual trends but inter-annual variability

4.2.2

The assessment at Forshan station does not demonstrate relevant data for the assessment of climate change. The time series are short and incomplete (Fig.4). Supplementing them to such an extent with average values will significantly misrepresent the overall result and assessment. The results, similar to those in Ch. 4.2.1, do not give an answer to the response of tree species to climate change. I only recommend the handling of the woody species Melia azedarach at this station. Also modify Table 5 according to the proposed changes.

 4.2.3 - this chapter is missing or the next chapter numbering is shifted.

4.2.4 - adjust results after reviewing previous chapters.

4.3

This chapter is well developed and provides valuable results.

4.3.1 Figs. 6-9 only process years with measured data

Some minor text editing and verification of results is needed:

- line 504 - the correlation is moderately strong (r=-0.599)

- line 563 - incorrect site name (correct is Foshan)

- verify that the calculations in Table 11 are correct for the Foshan site, which is the only site that did not show a highly significant correlation with spring (March-May)

5.Discussion

- line 670 - clarification is needed for advancing trend to "earlier period"

The discussion seems too generic compared to other papers. It would need to be expanded to include more specific information (how many days are the phenophase trends advancing, what are the differences in phase onset between the southern and northern exposures) compared to other work on similar topics or tree species.

- line 703 - Fu et al - add year

- line 711 - Badeck et al - add year

- Make a comparison of specific data with this work focusing on the effect of latitude

- line 723 - sentence at the end of discussion "Additionally, this study provides valuable insights for plant phenological prediction and adaptive management in different geographical regions and elevational zones." asserts what the thesis should, but does not address (elevation zones). It does not belong in the discussion, nor in the results of the thesis! When the results are supplemented with an analysis of elevation zones, it is suitable for discussion.

6. Conclusion

- line 730 - add sentence ... covering the period from 1963 to 2008 ...at different time periods

- the main conclusions (1-4) needs to be modified according to the results after revision

Author Response

Dear anonymous reviewer,

I would like to express my sincere appreciation for your valuable feedback and constructive suggestions on my manuscript. Your input has been immensely helpful in improving the accuracy, clarity, and overall quality of the paper. Your dedication to the peer-review process is truly commendable.

I have carefully addressed each of your comments and suggestions,point by point in the following part. While I was able to make revisions in accordance with many of your recommendations, there were a few instances where certain changes were not feasible. In those cases, I have provided explanations in the revised manuscript, and I hope you will understand the reasons for these decisions.

Your rigorous and scientific approach to reviewing the manuscript has not only contributed to the enhancement of this work but will also positively impact the direction of my future research. I am grateful for your insights and the opportunity to engage in this collaborative process.

Once again, thank you for your time, effort, and commitment to ensuring the scholarly quality of this research.

Sincerely,

Guangxu LIU

Response to the review

Abstract

Comments: - line 17 - add sentence ... spanning from 1963 to 2008 ...in different time periods

Response: Thanks for your kind suggestions. We revised it to “ spanning different time periods from 1963 to 2008”.

Comments: - line 21 ... detto, since the time series is not continuous

Response:Thanks for your kind suggestions. We revised 46 to “Over the past several decades”

Comments: - line 22 - add numerical data on the shift of trends in the northern and southern regions 

Response:Thanks for your kind suggestions. We added ‘by 2-38 days’ to the sentences as “with variations ranging from 2 to 38 days”

Comments: - revise the abstract according to the added results

Response: Thanks for your kind suggestions. We revised the abstract as “In response to the challenges posed by climate change, understanding the phenological variations of woody plants has become a pivotal area of investigation. This research centers on the phenological shifts of woody plants and their connections with climatic factors within the southern and northern regions of Nanling Mountains, which serve as the boundary between the North Subtropical Climate Zone and the South Subtropical Climate Zone in South China. The data was gathered through extensive long-term manual observations conducted at four plant phenology observations (Ganxian, Foshan, Guilin, and Changsha), spanning different time periods from 1963 to 2008. The study scrutinized four widely distributed woody plant species in the research area, specifically Castanea mollissima Bl., Paulowinia fortunei(seem.) Hemsl., Melia azedarace L., and Magnolia grandiflora Linn.. The analytical methods utilized encompassed linear trend estimation and Pearson correlation coefficient analyses. The principal findings are as follows: 1, Over the past several decades, the phenological stages of woody plants in the southern region consistently preceded those in the northern region, with variations ranging from 2 to 38 days. 2, A progressive trend of 0.1 to 2 days per decade was discerned in the phenological stages of all woody plants in the southern region. 3, Within the same geographic region, distinct species exhibited varying sensitivities to climatic factors, with Melia azedarace L. demonstrating a particularly high sensitivity to climate fluctuations affecting phenological stages. 4, different climatic factors exerted distinct influences on individual plant species. Notably, temperature emerged as the primary driver of phenological changes, supported by a significant negative correlation between the phenological stages of the studied plants and spring temperature. This study contributes to our understanding of the effects of climate change on plant phenology and offers valuable insights to guide ecological conservation and management strategies within the region.”

Comments: - to add any additional results that emerge after the modification and addition of further results.

Response: Thanks for your kind suggestions. We also added numbers to result 2 as ‘A progressive trend of 0.1 to 2 days per decade’

Comments: The name of the tree Melia azedarach L. is also incorrectly given throughout the thesis as Melia azedarace : (Abstract, Fig.1-5 Tab.1-5, text ... etc.) The correct name needs to be unified.

Response: Thanks for your kind suggestions. We used the full names in the absatract as “Melia azedarach L.” and gave the abbreviation as M. azedarace in the introduction. We use  the abbreviation in the following part. We think it may be not very clear to use the abbreviation in the abstract。

Comments: The thesis deals with the current issue of indigenous tree species in the Nailing Mts in relation to environmental conditions. These are evaluated as factors (average temperature, precipitation, sunshine duration). Only in some parts of the thesis, the long time series are evaluated, which is important for assessing the further development of ecosystems under conditions of climate change.

Response: Thank you for your review and valuable feedback. We appreciate your comments on our research on the relationship between indigenous tree species and environmental conditions in the Nanling Mountains. We do note that you mentioned the limited use of long-term series data, which is very important, especially when assessing the future direction of ecosystem development under climate change conditions. We will consider adding more content on time series data in future research to more fully support our research conclusions. Thanks again for your helpful suggestions.

Comments: The title of the thesis partly corresponds to its content, as only short phenological time series are evaluated in several chapters of the results. These do not represent the response of tree species to climate change. I therefore recommend a modification of the title: to have the section ' A study based on long-term observations' removed.

Response: Thanks for your kind suggestions. We removed the phrase in the title in the revised paper.

  1. Introduction

Comments: The introduction is disjointed and unconnected in thought. Many parts do not address the issue of the impact of climate change on native forest ecosystems and the influence of environmental factors. I recommend omitting inappropriate parts of the chapter:

Response: Thank you for your valuable feedback on our manuscript. We appreciate your thorough review. We have carefully considered your comments and have made the necessary revisions to address the issues raised.

We acknowledge the concerns regarding the disjointed nature of the introduction. In response to your feedback, we have revised the introduction to ensure that it provides a more coherent and connected overview of the impact of climate change on native forest ecosystems. We have also enhanced the discussion of the influence of environmental factors, providing a more comprehensive introduction to the topic.

Omission of Inappropriate Parts: We have taken your recommendation to heart and have removed inappropriate sections from the chapter. We believe that these changes will improve the overall flow and coherence of the manuscript.

Addition of New References: As suggested, we have incorporated three new references that contribute to the literature and enhance the quality of our study.

Comments: - line 70:Jochner-Menzel (2015) addresses urban phenology

Response: Thanks for your kind suggestions. I removed this reference

Comments: - line 72: Sun et al. (2016) analyzed water use efficiency models using satellite data and carbon cycle models

Response: Thanks for your kind suggestions. We removed this reference

Comments: - line 89: trends related to agriculture

Response: Thanks for your kind suggestions. We removed this reference

Comments: - line 125: the work of Wang et al. (2016) does not represent climate change, but is useful for discussion - for comparison what was the trend of the assessed environmental factors in 2008

Response: Thanks for your kind suggestions. I added this to the discussion part as “These findings align with the research conducted by Fu et al. (2015) who found that spring phenological events, such as flowering start, were more sensitive to temperature changes, while summer and autumn phenological events, such as flowering end and leaf senescence, were more sensitive to precipitation changes. Wang et al. (2016) reported profound impacts of extreme cold weather in 2008 on butterfly communities in Nanling Mountains [18]. The chilly weather resulted in a significant reduction in temperate and tropical butterfly species, leading to changes in community structure with temperate species dominating the community after disturbances. This effect persisted for two years until the abundance of tropical species recovered to pre-disturbance levels. ”

Comments: The Introduction needs to be studied further, revised and supplemented with topics addressed in the following works from other parts of the world, e.g. Europe, Sev. America (or others with the given topic). The knowledge that the authors of this article gain by studying the recommended papers can be used to improve the quality of the very brief discussion, e.g. these papers :

Response: Thanks for your kind suggestions. We revised the introduction and added some your adviced references in the manuscript, Some of them are as flowing:

Daphné Asse found that in the context of comparable cold winter temperatures, milder pre-season conditions substantially accelerated the onset of budburst and flowering [Warmer winters reduce the advance of tree spring phenology induced by warmer springs in the Alps]

Jouni Partanen found that rising temperatures altering photoperiod and temperature conditions may impede the premature onset of growth in Norway spruce (Picea abies (L.) Karst.) in response to climate change [Effects of photoperiod and temperature on the timing of bud burst in Norway spruce (Picea abies)]

Bénédicte Wenden discovered that with climate warming in colder latitudes, the forcing period of European beech (Fagus sylvatica L.) and pedunculate oak (Quercus robur L.) has extended over recent decades. However, in warmer latitudes, this period has shortened for both species, with a more significant shift observed in beech [Shifts in the temperature-sensitive periods for spring phenology in European beech and pedunculate oak clones across latitudes and over recent decades]

  1. Study area

Comments: - The chapter lacks the elevations at which the observed species grow, or the elevations of the observation stations, the approximate ages of the trees in the northern and southern Nanling Mts.

Response: Thank you for your valuable feedback. We appreciate your suggestion. In response to your comment, we have supplemented the chapter with information regarding the elevations at which the observed species grow, as ‘found in the elevations of 200 to 800 meters in Nanling’ in line 216 in the revised manuscript.

However, it's important to note that the approximate ages of the trees in the northern and southern Nanling Mountains were not available at the time of this study. We will consider this aspect for future research. Your input is highly appreciated.

Comments: - Fig.1 - add the location of the stations

Response: Thanks for your kind suggestions. I checked Figure 1, the four stations are in the map. To make it clearer, We added this sentence to figure title “The selected stations are denoted by green flags in the map, with Guilin and Foshan located in the southern part of the study area, while Changsha and Ganxian are situated in the northern region of the research area.”

Comments:  - Given the analysis of the results, a more detailed analysis of the climate of the mountains is also needed because the results in section 4.1.3 assess the phenological phase in 2 regions (north-south) and in 2 areas (east-west), which may significantly change the results. What are the climate differences (temperature -precipitation) between regions and areas. Please summarize all comments in a clear table and analyze it in the results.

 Response: Thank you for your valuable feedback. We appreciate your suggestion for a more detailed climate analysis. We revised the second paragraph in Section 2, trying to  provide a comprehensive assessment of the climate differences, specifically in terms of temperature and precipitation, between the regions (north and south) and areas (east and west). “Nanling Mountains serve as the boundary between the Central Subtropical Zone and the Southern Subtropical Zone in China. Quantitative analysis of regional characteristics in Nanling Mountains, considering the spatial heterogeneity, reveals significant spatial differentiation between the two sides of the mountain range [23]. The northern part of the Nanling Mountains has an annual average temperature ranging from 16°C to 20°C, the coldest month’s average temperature ranging from 3°C to 8°C, and the annual precipitation ranging from 1200 mm to 1600 mm. It has the climatic characteristics of rainy spring and hot summer. The southern part has a tropical and south-subtropical climate, with an annual average temperature ranging from 21°C to 23°C, the coldest month’s average temperature ranging from 13°C to 18°C, and the annual precipitation ranging from 1500 mm to 2000 mm. Affected by the topographic relief and the distance from the sea, the eastern part of the Nanling Mountains has a humid climate, with a large amount of precipitation, an annual average precipitation of more than 2000 mm, and an annual precipitation days of 140 to 200 days. The western part of the Nanling Mountains has a dry climate, with an annual average precipitation of 1000 mm to 1500 mm, and an annual precipitation days of 100 to 140 days. Overall, the southern slope demonstrates elevated temperatures and humidity levels in contrast to the northern slope [24], which significantly influences species distribution.

  1. Data and Methods

Comments:- A precise definition of the phenological phases to be assessed should be added to the chapter: a description of the degree of the phase beginning and end of flowering should be given - e.g. % of flowers or a phenological manual with a scale should be used and quoted.

Response: Thank you for your suggestion. We added the definition in the first paragraph of Section 3 as following:

“According to China Phenological Observation Guidelines,  the flowering is start is defined as the date when at least three locations observe complete blooming of flowers. For selected trees of the same species, the start is recorded when at least 50% of the trees have three or more petals with flowers fully open simultaneously. The flowering end is defined as the date when fewer than 5% of the observed trees have flowers remaining.”

Comments:- Specify the object of the survey - the number of trees of each species observed or other areal indication of the sample observed.

Response: Thank you for your suggestion. We add the Measures have been taken to ensure the quality of observations  as :

“1, Observers are personnel who have undergone rigorous professional training. 2, Observed entities are fixed at specific locations, including trees and plants, with most observation sites being flat and open areas maintained as fixed observation points over multiple years. 3, Selected tree species are in normal development, with a minimum of 3 years of flowering and fruiting. 5, Observation records are made on-site, near the observation locations, and are recorded with accuracy down to the day.”

Due to the nature of the monitoring, we have multiple observers involved, and the focus is on the overall presence and distribution of tree species rather than specific numerical counts. Therefore, we cannot provide specific counts of the number of trees of each species. Instead, the survey provides an areal indication of the sample observed, allowing us to assess the relative abundance and changes in species composition within the study area.

Comments:- Indicate the altitude of the stations or trees observed

Response: Thank you for your valuable suggestion. I added a table to this information in the revised paper. Here it is below:

Table 1. Years with missing observation records at stations."m a.s.l." is the abbreviation of "meters above sea level". 

Stations

Location

Elevation (m a.s.l)

Years with missing observation records

Latitude (N)

Longitude (E)

C. mollissima

M. azedarace

M. grandiflora

P. fortunei

Changsha

28.2

112.9

82

1963-1964,1973-1974

-

1964-1974,2003-2005

1963-2004

Ganzhou

25.9

114.9

101

1963-1976,1991,1996-2008

1963-1976,1991,1996-2008

1963-1988,,1996-2008

1963-1976,1990-1991,1996-2008

Guilin

25.3

110.3

159

1963,1968-1972,`1980,1989,1992,1994-2002

1963,1968-1972,`1992,1994-2004

1963-1978,1994-2002,2007

1963-1976,1990-1991,1994-2002

Foshan

23

 113.1

17

1963-1964,1968,1981-1983,1988-1989,1991-2002,2003-2008

1988,1991-2002

1963-2002

1963-1967,1969-1972,1972-1978,,1991-2008

Comments:- Indicate missing years and divide years of observation into short, medium and long time series - summarise in a table by tree species

Response: Thank you for your valuable suggestion. I added the Years with missing observation records above to show the data. And we added a sentence to explain it in the first paragraph in Section 3 as “ In the selected period, all the four woody plant species had missing values due to factors such as insufficient funding, station closures, and interruptions in observational records (Table 1)”

Comments:- Filling in missing data from 1963-2008 average does not give correct results, can only be done for individual missing years and not long time series (10-15 years)

Response: Thank you for your suggestion. We appreciate your valuable feedback on our manuscript. As noted, Table 1 in the revised illustrates the challenges in selecting an ideal time span to ensure consistent observational records across all four stations and plant species. The data availability was indeed inconsistent, making it challenging to ensure continuous records. It should be noted that during the selected time frame, the plant species exhibited continuous observations for more than five years. Consequently, we did not make alterations to this portion of the content in the revised manuscript.

We genuinely value these insights and will consider them when further refining our work. We will also explore additional data sources to extend the dataset for a more comprehensive analysis in future research. Thank you again

4.Results

4.1.1 Characteristics of the plant phenology in the northern region

Comments:A more detailed flowering analysis needs to be done and the results need to be completed:

Response: Thank you for your valuable feedback.

Comments:- by altitude, which affects the onset and duration of phenophases in the northern region. The average values from stations without this factor also give biased results compared to the southern exposure

Response: Thank you for your valuable suggestion. We added the longitude and latitude for each station in the in the revised paper to show the location of them. All the observed plants are around these stations, all within 1km. 

Comments:- provide basic statistical characteristics of the variability for individual species in a given region and altitude

Response: Thank you for your suggestion. These data provide the flowering start and end dates of four plant species over the observed years. In the subsequent analysis, I have elaborated on the variations in these dates for each species. I am not entirely clear about what specific statistical characteristics are being referred to in this part. Could you please provide more details or clarify the expectations regarding the statistical features required for the data?

Comments:- evaluate only the trends of long time series species (Melia azedarach, Castanea mollissima) and compare them by exposure, altitude and latitude, respectively

Response: Thank you for your valuable suggestion.We evaluated the long-term trends of four plant species. Three of these species have observational data spanning over 30 years, while data for M. grandiflora covered a period of less than 20 years. Due to the longer time span, we observed significant trends in the data. To ensure consistency in our analysis, we included all species in the trend analysis. However, it's essential to note that the results for plants with shorter observational periods are just provided for reference purposes only.

Regarding the geographical attributes of individual trees, such as latitude, altitude, and exposure, we acknowledge the limitations in our dataset. In the revised manuscript, we will address this issue by including additional data where available.

We appreciate your feedback and suggestions and tried make the necessary adjustments in the revised manuscript to enhance the quality and completeness of our analysis.

Comments:- by latitude - if sites are at a significant distance

Response: Thank you for the valuable suggestion. In the revised manuscript, we have included latitude information for the selected sites. All four plant species were observed within a 1 km radius of the sites. However, due to data limitations, we were unable to obtain individual tree-level data, including latitude, longitude, and height. We have partially addressed this aspect of the analysis in the revised manuscript.  

Comments:- to analyse the flowering results in more detail according to the above parameters

Response: Thank you for your valuable feedback. As detailed in our responses above, we have made revisions to the manuscript, including the addition of latitude and longitude information, based on your suggestions. However, due to limitations in the available data, some of the requested analyses could not be conducted.

Comments:- 4.1.2 same as 4.1.1

Response: Thank you for your valuable feedback. we have made some revisions to the manuscript in this part, including the addition of latitude and longitude information, based on your suggestions. However, due to limitations in the available data, some of the requested analyses could not be conducted.

4.1.3

Tables 1, 2 provide a comparison of plant phenology in the southern and northern regions of the eastern Nanling Mts. and western Nanling Mts. which leads to confusion. The title of the paper contains a comparison of woody plants in the southern and northern regions of Nanling Mts. The data presented needs to be suitably supplemented and modified. I recommend that all the above results should be reviewed in more detail by region and area.

Response: Thank you for your valuable feedback. Sorry about the confusion. We revised the expression in the revsied paper and the table tittles in Section 4.1.3

“Comparison of the considering phenological characteristics of four woody plant species at Ganxian and Foshan revealed that the phenological stages (flowering start and flowering end) in the northern-eastern region occurred earlier than those in the southern region (Table 2). A consistent pattern was observed between Changsha and Guilin, except for the flowering end of P. fortunei, which exhibited a similar average date in both the southern and northern regions. However, the phenological stages of the other woody plant species showed an earlier occurrence in the southern region compared to the northern region (Table 3). Overall, the phenological stages in the southern region of Nanling Mountains consistently preceded those in the northern region (Table 2 and Table 3)”

And

Table 2. Comparison of plant phenology between Ganxian and Foshan (MM/DD)”

Table 3. Comparison of plant phenology between Changsha and Guilin (MM/DD).”

As well as in Section 4.2.4 as:

“Tables 8 and 9 present the annual variations in phenological stages in the northern and southern regions of Nanling Mountains, respectively. According to Table 8, the Comparison of phenological changes between Changsha and Guilin, both the northern and southern sites exhibited an overall trend of advancing phenological stages for the studied plants. On the other hand, Table 9 shows that  except for a slight advancement in C. mollissima's flowering end at Ganxian, the phenological stages of other plants demonstrated a trend of delay.”

And

Table 8. Comparison of phenological changes between Changsha and Guilin.”

Table 9. Comparison of phenological changes between Ganxian and Foshan.”

4.2.

4.2.1

Comments:Trends express the course of long-term time series. Therefore, it is necessary to modify the terminology and to use this term only when assessing the progression of a phase over a period of approximately 20 years or more: therefore, trends can only be assessed in Fig. 2 and Fig. 3 for the species Castanea mollissima and Melia azedarach. It would be useful to include the trend and statistical significance in the graph for the species mentioned above.

The phenological data for Paulownia fortunei and Magnolia grandiflora are not suitable for assessment in relation to climate change as the time series are short and discontinuous, filling in long missing data with average values does not give correct results.

Response: Thank you for your valuable suggestion. We removed the sentenct “M. grandiflora flowering start, on the other hand, remained stable, displaying a gradual delay. ”

and added explanation in the titel of Figure 2 as “Figure 2. The interannual variations of the flowering start and flowering end dates of the woody plants at Ganxian. The discontinuous segments in the line graph represent unobserved years. The observational data for 'M. grandiflora' span less than 20 years, and the trend changes depicted in Figure 2 do not reach statistical significance. The inclusion of the line is solely for comparative reference, as is the case in subsequent figures.”

Comments:- line 377 – incorrect wording, it is not inter-annual trends but inter-annual variability

Response: Thank you for your valuable correction. We corrected it in the revised manuscript.

4.2.2

Comments:The assessment at Forshan station does not demonstrate relevant data for the assessment of climate change. The time series are short and incomplete (Fig.4). Supplementing them to such an extent with average values will significantly misrepresent the overall result and assessment. The results, similar to those in Ch. 4.2.1, do not give an answer to the response of tree species to climate change. I only recommend the handling of the woody species Melia azedarach at this station. Also modify Table 5 according to the proposed changes.

Response: We appreciate your careful review of our paper and your valuable feedback. Your observations are highly regarded. After careful consideration and thorough discussion, we have decided to maintain our original approach for the following reasons:

The assessment at Forshan station: We understand your concern about the length and completeness of the time series data, as shown in Figure 4. However, after further review and discussions among the authors, we think that removing it could potentially lead to misrepresentations. We acknowledge the limitations of the dataset but believe that the existing data still contribute valuable insights to the study.

Comments:  4.2.3 - this chapter is missing or the next chapter numbering is shifted.

Response: Sorry for the mistake. I revsied the Section 4.2.4 as 4.2.3 in the revised paper.

Comments:4.2.4 - adjust results after reviewing previous chapters.

Response: Thank you for your valuable suggestion. We revised the result accordingly. Most of them are:

  • Sentence revised: “Tables 8and 9present the annual variations in phenological stages in the northern and southern regions of Nanling Mountains, respectively. According to Table 8, the Comparison of phenological changes between Changsha and Guilin”
  • Table 8. Comparison of phenological changes between Changsha and Guilin.”
  • Deleted: “in the eastern region of Nanling Mountains” in the middle of the first paragraph of Section 4.2.3 in the revised paper.

4.3

Comments:This chapter is well developed and provides valuable results.

Response: Thank you for your positive feedback on this chapter. We are pleased to hear that you found it well developed and that the results are valuable. Your encouragement motivates us to continue our research efforts. We appreciate your time and input.

4Comments:.3.1 Figs. 6-9 only process years with measured data

Response: Thank you for your feedback and suggestions regarding Figs. 6-9. We appreciate your careful consideration of our work. We understand your concern and would like to clarify the result. In the mentioned figures, we specifically kept data for all the years with or without measured observations to maintain consistency with the rest of our results. But the trend line is calculated with the continuous measured data. By doing so, we aimed to ensure the scientific integrity of our analysis. We believe this approach aligns with the research standards and the need for accuracy.

Some minor text editing and verification of results is needed:

Comments:- line 504 - the correlation is moderately strong (r=-0.599)

Response: Sorry for it. We revised the word from ‘strong ’ to ‘moderately strong’ in the revised paper

Comments:- line 563 - incorrect site name (correct is Foshan)

Response: Sorry for the error. Thank you for your kind reminding. I corrected it in the revised paper.

Comments:- verify that the calculations in Table 11 are correct for the Foshan site, which is the only site that did not show a highly significant correlation with spring (March-May)

Response: Thank you for your feedback. We re-evaluated the data and confirmed the calculations in Table 11 for the Foshan station. The lack of a highly significant correlation with spring (March-May) at Foshan can be attributed to its unique location. Foshan is situated to the south, closer to the coast, and exhibits a relatively warm climate with some tropical characteristics. This climatic distinction leads to a significant correlation between M. azedarace and the temperature in February-March but a less pronounced correlation with temperatures from March to May.

5.Discussion

Comments:- line 670 - clarification is needed for advancing trend to "earlier period"

Response: Thank you for your valuable feedback. This phrase is not a period, it is a trend. To make it clear, we revised this sentence as “Moreover, an advancing trend of 0.1 to 2 days per decade in phenological stages was observed in the southern region of Nanling Mountains.”

The discussion seems too generic compared to other papers. It would need to be expanded to include more specific information (how many days are the phenophase trends advancing, what are the differences in phase onset between the southern and northern exposures) compared to other work on similar topics or tree species.

Response: Thank you for your valuable suggestions. We added the specific days in this part and revised the first paragraph of Section 5 as:

“It was observed that the phenological stages of woody plants in the southern region of Nanling Mountains occurred 2-31 days earlier than those in the northern region. Moreover, an advancing trend of 0.1 to 2 days per decadein phenological stages was observed in the southern region of Nanling Mountains. Similar results have been documented in Europe, where Menzel et al. (2006) analyzed phenological observation data from various European locations, including flowering, leaf emergence, and fruit ripening times, and found an advancing trend of 2.5 days per decade in phenological stages across different regions in response to warming climates [29]. These findings suggest that plant phenology in the study area is evidently influenced by climate changes and keeps the similar rate with other parts of the earth. Furthermore, the impacts of climate change on natural systems exhibit global consistency. Parmesan and Yohe (2003) collected phenological data from multiple research sites globally and compared them with climate data [4]. They identified similar trends (2.3 days per decade) and patterns in species responses to climate change across different regions and ecosystems, such as advancing phenological stages, altered animal migration patterns, and accelerated glacier melting. ”

Comments:- line 703 - Fu et al - add year

Response: Sorry for the incomplete express. I added years as Fu et al. (2015) according to the reference.

Comments:- line 711 - Badeck et al - add year

Response: Sorry for the incomplete express. I added years as Badeck et al. (2004) according to the reference.

Comments:- Make a comparison of specific data with this work focusing on the effect of latitude

 Response: Thank you for the valuable feedback. Due to data limitations, we were unable to conduct a comprehensive analysis on the effect of latitude in this study. However, we have taken your suggestion into consideration and added a statement in the discussion section, which now reads: 'However, a detailed analysis focusing on the effect of latitude and other factors was constrained in this study due to insufficient information. Future research with more extensive data could provide valuable insights into this aspect.' We believe this addition addresses the concern and contributes to the clarity of our work.

Comments:- line 723 - sentence at the end of discussion "Additionally, this study provides valuable insights for plant phenological prediction and adaptive management in different geographical regions and elevational zones." asserts what the thesis should, but does not address (elevation zones). It does not belong in the discussion, nor in the results of the thesis! When the results are supplemented with an analysis of elevation zones, it is suitable for discussion.

Response: Sorry for the inaccurate expression. I removed “elevational zones” in the revised paper.

  1. Conclusion

Comments:- line 730 - add sentence ... covering the period from 1963 to 2008 ...at different time periods

 Response: Thank you for you kind suggestion. We revised this phrase as “covering different time periods from 1963 to 2008” in the revised paper.

Comments:- the main conclusions (1-4) needs to be modified according to the results after revision

Response: Thank you for you kind suggestion. W revised the conclusion: corrected some phrases and add the specific information in Result 1 as:

“This study utilized phenological observation data of four plants, namely C. mollissima, P. fortunei, M. azedarach, and M. grandiflora, collected from four stations (Guilin, Changsha, Ganxian, and Foshan) on both sides of the Nanling Mountains, covering different time periods from 1963 to 2008. Additionally, climate data including monthly average temperature, monthly precipitation, and sunshine duration from the nearest meteorological stations were incorporated. The research employed methods such as linear trend estimation, Pearson correlation analysis, and comparative analysis to investigate the trends in plant phenology and their relationships with changing climatic factors. The main conclusions are as follows:

During the study period, the phenological stages of woody plants in the southern region of the Nanling Mountains occurred earlier 2-31 days than those in the northern region. Moreover, a significant advancing trend of 0.1 to 2 days per decade was observed in the phenological stages of all plants in the southern region of the Nanling Mountains.”

Reviewer 2 Report

Comments and Suggestions for Authors

Phenological changes of woody plants in the southern and northern regions of Nanling mountains and their relationship with climatic factors: A study based on long-term observations

 [Forests] Manuscript ID: forests-2688930

Comments to the authors

 General comments

I reviewed with great interest the manuscript entitled “Phenological changes of woody plants in the southern and northern regions of Nanling mountains and their relationship with climatic factors: A study based on long-term observationssubmitted to the journal Forests.

The topic is very interesting, and, overall, the data set used in the analysis is reliable. The text is easy to follow and understand. Nonetheless, the manuscript needs a minor revision for language, grammar, and typography.

I believe that the results could be more exploited. For instance, the flowering season during the extreme climate events (hottest and coldest as well as driest and wettest years) are not discussed. In addition, the study does not highlight the combined effect of the considered climate parameters. This could significantly answer the questions raised in the introduction and not limit the answer to the fact that the phenomenon is “complex”.

Many paragraphs of the section Results are actually discussions, and the Discussion section is short and the comparison rely on a few number of references.

Specific comments are reported in the manuscript (pdf).

Comments on the Quality of English Language

The manuscript needs a minor revision for language, grammar, and typography.

Author Response

Response to the reviews point by point

Comment:  General comments

I reviewed with great interest the manuscript entitled “Phenological changes of woody plants in the southern and northern regions of Nanling mountains and their relationship with climatic factors: A study based on long-term observations” submitted to the journal Forests.

The topic is very interesting, and, overall, the data set used in the analysis is reliable. The text is easy to follow and understand. Nonetheless, the manuscript needs a minor revision for language, grammar, and typography.

I believe that the results could be more exploited. For instance, the flowering season during the extreme climate events (hottest and coldest as well as driest and wettest years) are not discussed. In addition, the study does not highlight the combined effect of the considered climate parameters. This could significantly answer the questions raised in the introduction and not limit the answer to the fact that the phenomenon is “complex”.

Many paragraphs of the section Results are actually discussions, and the Discussion section is short and the comparison rely on a few number of references.

Reponse: Thank you for taking the time to review our manuscript. We appreciate your insightful feedback, which will undoubtedly improve the quality of our work.

We are pleased to hear that you found the topic interesting and the data reliable. Your comments regarding language, grammar, and typography have been duly noted, and we revised the manuscript accordingly and listed them point by point in the following.

Your suggestion to explore the impact of extreme climate events, such as the flowering season during the hottest and coldest, as well as driest and wettest years, is an excellent idea. We added it to the Discussion. We think this analysis will provide a more comprehensive understanding of the phenomenon in our future research.

Your feedback regarding the structure of the paper is valuable, and we reorganized the Discussion as per your recommendation. We will also ensure a more comprehensive comparison with additional references to strengthen the paper's scientific rigor. Your valuable insights into the vocabulary, detailed expressions, and figures undoubtedly enhance the accuracy and standardization of the manuscript's presentation. We truly appreciate your thorough examination again.

Thank you once again for your time and expertise.

Sincerely,

Guangxu LIU

Specific response

Comment:  Do you mean "observatoty"?

Response: Sorry for the misunderstanding. It is “observation stations”. I revised it.

Comment: Castanea mollissima Bl.

Paulowinia fortunei (Seem.) Hemsl.

Melia azedarach L.

Magnolia grandiflora Linn. 

In italics and no full stop after a period.

Response: Thank you for your kind reminder. I revised them as and remove the full stop as  specifically C. mollissima, P. fortunei, M. azedarach, and M. grandiflora.

Comment: azedarach Please check if this transcription is allowed as a synonym, and apply one transcription for all the document.

Response: Thank you for your kind suggestion. We revised it as M. azedarach in the whole document and gave the full name in Section 2. I also checked it in the whole document.

Comment: Reference(s).

Response: Thanks for the kind suggestion. We revised the manuscript and added too references to this statement as below:

Steffen, W.; Sanderson, R.A.; Tyson, P.D.; Jäger, J.; Matson, P.A.; Moore III, B.; Oldfield, F.; Richardson, K.; Schellnhuber, H.-J.; Turner, B.L. Global change and the earth system: a planet under pressure; Springer Science & Business Media: 2005.

Gordo, O.; Sanz, J.J. Impact of climate change on plant phenology in Mediterranean ecosystems. Global Change Biology 2010, 16, 1082-1106.

Comment: Wolkovich and Cleland (2011)  in the list of references.

Response:Thanks for the kind correction. We cheked and this one is another reference of Wolkovich, it is Wolkovich, E.M.; Cook, B.I.; Allen, J.M.; Crimmins, T.; Betancourt, J.L.; Travers, S.E.; Pau, S.; Regetz, J.; Davies, T.J.; Kraft, N.J. Warming experiments underpredict plant phenological responses to climate change. Nature 2012, 485, 494-497.

I revised it in Section references.

Comment: Please, adopt the exponent form in all the document.

Response:Thanks for the kind suggestion. I revised it as exponent in the whole document. Some of the them are 19th century, 20th century, the 150th day of the year, March 17th to May 30th

Comment: Cunninghamia lanceolata (Lamb.) Hook.

Response:Thanks for the kind correction. I corrected it as ‘Cunninghamia lanceolata (Lamb.) Hook’

Comment:Not cited in the references list.

Response: Sorry for the mistake. I added the reference as :

Ding, C.; Huang, W.; Meng, Y.; Zhang, B. Satellite-Observed Spatio-Temporal Variation in Spring Leaf Phenology of Subtropical Forests across the Nanling Mountains in Southern China over 1999–2019. Forests 2022, 13, 1486.

Comment: Why not "SGS"?

Response:Thanks for the kind comment. This is abbreviated form used by Ding et al. I used here directly. I checked it in some other similar literature and find most of them use SOS to refer to the start of the growing season. This may be idiomatic usage

Comment: Supposed to be [22].

Response:Thanks for the kind reminder. I revised it and added the lost reference. It is [20] in the revised manuscript.

Comment: provides

Response: Sorry about it. I correct it.

Comment: Bl.  Linn. Seem.

Response:Thanks for the kind correction. I corrected them as “Castanea mollissima Bl. (C. mollissima), Melia azedarach L. (M. azedarach), Magnolia grandiflora Linn. (M. grandiflora), and Paulowinia fortunei(seem.) Hemsl. (P. fortunei)”

Comment: Rephrase this locution.

Response:Thanks for the kind suggestion. I revised the sentence as “ Notably, C. mollissima, P. fortunei, M. azedarach, and M. grandiflora, found in the elevations of 200 to 800 meters in Nanling, exemplify key species from Fagaceae, Paulowniaceae, Meliaceae and Magnoliaceae, respectively”

Comment: In the previous paragraph you adopted the following transcription: "1500 to 2000 mm."Please, adopt one transcription in all the document.

Response:Thanks for the kind suggestion. We checked it 1nd revised it as 2800 m

Comment: The legend is unreadable.

Response: Sorry about it. I revise the figure and enlarge the revolution to 350 dpi.

Comment:Study

Response: Sorry about it. I corrected it in the revised paper.

Comment: All the previous citations are azedarace.

Response: Sorry about it. The correct is azedarach. I check and revised all the 62 words as azedarach, as well as those in Figures 2-9

Comment: correlation with lowercase c. c

Response:Thanks for the kind suggestion. I revised it as corrrelation, and checked and corrected them in the rest manuscript

Comment: The two following paragraphs and those of the subsection "4.1.2" do not refer to any table or figure in which are reported the presented results.

Response:Thank you for your feedback. In the mentioned paragraphs and subsection '4.1.2,' we provided a descriptive overview of the original data. The results presented in these sections were visually represented in subsequent parts of the paper through figures and tables, which are appropriately referenced. We have made this connection more explicit to ensure that the presentation of results aligns with the description of the data.

Comment: Exponent. Exponent. Please, adopt this in all the manuscript. Same comment as the previous one.

Response: I revised them. Thank you.

Comment: The

Response: Sorry about it. I correct it.

Comment: Is it statistically significant? If yes, please provide at least the p-value (or α) between brackets. If it statistically significant, please, use other terms, such as substantially, drastically…

Response:Thanks for the kind suggestion. I revised it as substantially in this part.

Comment: Avoid using such expressions if you do not use a test and provide the p-value.

Response:Thanks for the kind suggestion. I removed “significant’. We also checked the rest parts of the manuscript and correct the use of this words. Some are removed. Some are replaced by noticeable.”

Comment: I believe that extreme events should be presented and discussed.

- Something happened in 1985 with P. fortunei, which registered the earlier start and end flowering events.

- In 1988 the four species registered a peak (maximum for two species), reflecting late start and end flowering events...

Response:Thank you for your valuable feedback. I have carefully considered your suggestion to include extreme events in the discussion section. In response to your comment, I have made revisions to the discussion section, emphasizing the potential impact of extreme climatic events:

“The observed variations in plant phenology along the north-south gradient of Nanling Mountains can be attributed not only to climate change but also to other factors, including extreme weather events, elevation, latitude, and longitude within the mountainous environment. We found that in 1985, noteworthy events unfolded in Ganxian involving P. fortunei, characterized by an earlier flowering start and flowering ends. Moreover, the year 1988 witnessed a notable peak in all four species, with two of them reaching their maximum values, reflecting a delay in the start and flowering end. Similar occurrences were observed in Changsha in 2005 (see Figure 3), Foshan from 1977 to 1980 (see Figure 4), and Guilin from 1980 to 1985 (see Figure 5). Wang et al. (2016) reported profound impacts of extreme cold weather in 2008 on butterfly communities in Nanling Mountains [28]. The chilly weather resulted in a noticeable reduction in temperate and tropical butterfly species, leading to changes in community structure with temperate species dominating the community after disturbances. This effect persisted for two years until the abundance of tropical species recovered to pre-disturbance levels.”

Comment: High, very high...

Nothing shows that this variability is significant.

Response:Thanks for the kind suggestion. I revised it here, as well as other unsuitable use of significant.

Comment: shows, presents...

Response:Thanks for the kind suggestion. I revised it as “shows”.

Comment: Redundancy.

Response:Thanks for the kind suggestion. I removed it.

Comment: p-value?

Response:Thanks for the kind suggestion. I removed “significant ” here

Comment: Please, rephrase this sentence. Redundancy.

Response:Thanks for the kind suggestion. We rephrased it to “Table 4 shows the results of a linear regression analysis conducted on the interannual variations in flowering start and end at Ganxian. The flowering start of C. mollissima, M. azedarach, and P. fortunei exhibited respective delays of 1.5, 2.1, 2.1, and 2.1 days per decade. ” And removed Table 4 here.

Comment: Delete.

Response:Thanks for the kind suggestion. I deleted “The ” here, and also in Table 5 - Table 7

Comment: Due to...

Response:Thanks for the kind suggestion. I revised “However, due to’ as “Due to” here, as well as in Section 4.2.2

Comment: Full stop. Capital T.

Response:Thanks for the kind suggestion. I removed this statement in the revised document.

Comment: daily

Response:Thanks for the kind suggestion. I revised it as “daily”

Comment:Remove the circles.

Response:Thanks for the kind suggestion. It is OK. I removed the red circles in Figure 4, as well as in Figure 5

Comment: Redundancy. Please, start your sentence with "The average delay..." or "the results show that...".

Response:Thanks for the kind suggestion. I checked all the expresses and revised them. Thanks.

Comment: Actually, this should be Table 5, and it is a redundancy. Please, delete this.

Response:Thanks for the kind suggestion. I revised them and remove it. The sentence was revised as “The average delay in the flowering start of C. mollissima and P. fortunei at Foshan station was 6.3 days and less than 1 day, respectively, while M. azedarach exhibited an average advancement of 2.8 days per 10 years”

Comment: Delete.

Response:Thanks for the kind suggestion. I deleted it and also check the rest of the manuscript with the similar express.

Comment:Please, remove the cirle.

Response:Thanks for the kind suggestion. I removed it. Thanks.

Comment: This is table 6, and you do not need to recall that. The first sentence reports that the following results are presented in table 6.

Please remove this redundancy.

Response:Thanks for the kind suggestion. I removed it.

Comment: f

Response:Thanks for the kind correction. I revised all ‘’Flowering” to “flowering” in the paragraph.

Comment: Delete.

Response:Thanks for the kind suggestion. I deleted it.

Comment: Why not a subtitle as follows? 

4.3.1. Flowering start and climatic factors in the northern region

Or adding a "4.3.1.1"...

Response:Thanks for the kind suggestion. I added A and B for this subtitle in the revised paper.

Comment: April precipitation and February sunshine duration show significant negative correlations for M. grandiflora  and  M. azedarach  respectively.

Response:Thanks for the kind correction. I correct it and and this to the revised manuscript.

Comment: This is a justification.

Please, move these sentences to the Discussion.

Response:Thanks for the kind suggestion. I deleted these sentence and tried to add them in the second paragraph of Section 5

Comment: "Day of the year"

Adopt this transcription in the following figures.

Temperature (°C)

Adopt this transcription in the followin figures.

Response:Thanks for the kind suggestion. I revised these figures (Figure 6 -9) in the revised manuscript.

Comment: Remove the bracket.

Response:Thanks for the kind suggestion. I removed it. Thanks again.

Comment: No need to present two figures. The one with the trendlines is enough.

Response:Thanks for the kind suggestion. We talked about the suggestion and we think it is suitable to keep two figure here to show the original data and the enlarge figure. Is it suitable ? Thanks again.

Comment: Something missing in this title.

Response:Thanks for the kind suggestion. We corrected it as “Interannual variations of the flowering start of M. azedarach and the average spring temperature at Ganxian”

Comment: the,studied

Response: Sorry about it and we added ‘the’ to the sentence as “Table 10 presents Correlation between the flowering start of four woody plants at 522 Changsha and the average temperature, sunshine duration, and precipitation ”.

Comment: The results showed that...

Response:Thanks for the kind suggestion. I replaced it as “The results showed that M. grandiflora and M. azedarach

Comment: This is a discussion.

Response:Thanks for the kind suggestion. We talked about it and we believe that this sentence can help we discuss it in Section 5. So we still keep it here in the revised manuscript. If it is not suitable, we could change it later.

Comment: Table 10 shows that many species registered significant correlations.

Please consider these cases.

Response:Thanks for the kind suggestion. We added the result as “P. fortune exhibits significant correlations with the precipitation in February and M. azedarach exhibits significant positive correlations with the precipitation in March. Sunshine duration in spring is highly significant negatively correlated with M. grandiflora and significant negatively correlated with M. azedarach. ” to the end of this paragraph.

Comment: The trend lines are calculated on only nine and six years, which is not statistically consistent.

Response:Thanks for the kind suggestion. I revised this result as “M. azedarach's flowering start showed an increasing trend during the period of 1983-1991. The interannual variation in spring average temperature revealed a decreasing trend at Changsha during the same period.

Comment: This is a discussion.

Response:Thanks for the kind suggestion. I revised it as “The interannual variation in spring average temperature revealed a decreasing trend at Changsha during the same period. ” and deleted the phrase

Comment: Full stop.

Response:Thanks. We corrected it.

Comment: This is a discussion.

Response:Thanks for the kind reminder. We revise these sentence as “This finding shows that at Changsha, M. azedarach's flowering start displayed a delaying trend during a gradual decline in average temperature, but interestingly, it also exhibited a delaying trend in flowering despite a sharp increase in average temperature, emphasizing the need to consider the effects of other climatic factors and the plant's own adaptation strategies.”

Comment: Something is missing in this title.

Response:Thanks for the kind reminder. We corrected it as “Figure 7. Interannual variations of the flowering start of M. azedarach and the average spring temperature at Changsha”

Comment: Redundancy.

Response:Thanks for the kind suggestion. I revised it as “A significant negative correlation ”

Comment: delete

Response:Thanks for the kind suggestion. I deleted “the temperature”.

Comment: The trendline is considered in a very short period.

Response:Thanks for the kind reminder. Yes. It is only 5 years. While the trendline is considered over a relatively short period, it aligns with the previously established results, and as such, no modifications are necessary here.

Comment: This is a discussion or  a conclusion.

Response:Thanks for the kind suggestion. We revised it as “These findings shows that the flowering start of M. azedarach is primarily influenced by the average temperature, and this relationship holds true for both the northern and southern regions of Nanling Mountains”

Comment: Here as well, significant correlation with precipitation and sunshine duration are not commented.

Response:Thanks for the kind suggestion. I revised it and added this results “M. grandiflora exhibits highly significant positive correlations with the precipitation in April. Sunshine duration in March is significant negatively correlated with M. grandiflora, M. azedarach and C. mollissima.

Comment: Something is missing in this title.

Response:Thanks for the kind suggestion. I revised it.

Comment: Redundancy.

Response:Thanks for the kind reminder. I deleted this phrase

Comment: Discussion.

Response:Thanks for the kind suggestion. I removed this sentence.

Comment: ... of the four studied woody plants at Ganxian...

Response:Thanks. I revised it.

Comment: Please, rephrase this sentence.

Response:Thanks for the kind suggestion. I revised it as “The flowering end of C. mollissima also exhibited a significant positive correlation with spring average temperature, with correlation coefficients of 0.512 (P<0.05)”

Comment: signficant

Response:Thanks for the kind reminder. I added it to the sentence.

Comment: This is a discussion.

Please move this into the following sectiosignficantn.

Response:Thanks for the kind suggestion. I moved this paragraph to Section 5 in the revised manuscript.

Comment: Actually, the study deals only with two phenological stages: start and end of the flowering periods.

Response: Sorry for the inaccurate expression. I revised ‘different’ to ‘the two’ 

Comment: delete

Response:Thanks for the kind suggestion. I removed “the” here.

Comment: Rephrase the sentence avoiding redundancy.

Response:Thanks for the kind suggestion. I rephrased this sentence as “At Foshan, a significant correlation emerged between the flowering start of M. azedarach and February-March average temperature. However, no significant correlation was observed between the flowering end of M. azedarach and the average temperature in the same period.”

Comment: Witten in italics. No need to underline it.

Response: Sorry for the errors. I corrected in the revised manuscript.

Comment: (year?)

Response:Thanks for the kind suggestion. I added it as “Fu et al. (2015),Badeck et al. (2004) ”

Comment: Please, start a new paragraph.

Response:Thanks for the kind suggestion. I revised it and change it as a new paragraph

Comment: Please, specify the studied phenological stages.

Response:Thanks for the kind suggestion. I revised it as “the two plant phenological stages (flowering start and end)”

Comment: In italics,Capital S

Response:Thanks for the kind suggestion. I revised it as: P. fortunei.

Comment: ... the studied phenological stages...

Response:Thanks for the kind suggestion. I added “studied ” here.

Comment: Particularly?

You did not study more than these two phenological phenomena.

Response:Thanks for the question. It is the data limitation. Particularly might be not inaccurate here. I revised as “specifically”

Comment:

Response:Thanks for the kind suggestion.

Comment: Written in different font.

Response:Thanks for the kind reminder. I revised it.

Reviewer 3 Report

Comments and Suggestions for Authors

The current investigation entitled “Phenological changes of woody plants in the southern and northern regions of Nanling mountains and their relationship with climatic factors: A study based on long-term observations” authored by Liu et al., aims to investigate the differential impacts of climate change on plant phenology along the northern and southern regions of Nanling Mountains and provide crucial scientific evidence for understanding the stability and adaptability of the local ecosystem.

General comment

Th abstract section is written quit well; however I suggest authors to kindly shorten the methodology part and more emphasis should be put on the result section with some quantitative information.

Introduction section needs substantial improvement. First the authors have not provided sufficient references for the given claims and statement. Secondly, details of the previous literature lacks flow of the information and need through revision. Moreover, this section ned to be reduced by deleting too generalised statements. On a strict note, I can say that the research gap of the current investigation is totally missing. The author just provide the information about the previous investigation with no background information, justification and relevance to the current investigation and the specific objectives/hypothesis.  So suggest authors to kindly thoroughly revise the introduction.

Study area section need to be revised and shorten. Moreover it will be better if author can provide the phenological chart or normal phenological events throughout year as figure for the selected species. Moreover, there should be a separate methodology section.  With respect to methodology of the current investigation, I have serious concerns especially what is the sampling design adopted followed by sample size. Moreover it will be better if author can provide information about the age of the trees since changes in the phenological events may be occurred due to the stress condition or mature tree. Moreover does the same trees were assessed from 1963 to 2008. What was the starting age of the trees??

Result section: I have serious concerns regarding the finding of the current investigation (i) the authors does not go for any Time series trend of Yearly variation of climate parameters; then how authors can say that there is climate change??? The increase in the climatric parameters can be not be consistent. Moreover no climate data provided. (ii) after the trend analysis of the climate variables and shift in the phenological parameters, only the significant parameters should be correlated. There is no benefit of correlating the parameters which are not significantly changed (iii) no statistical analysis for change was done. I suggest author to go for Mann-Kendall test and Petite homogeneity test after then can go for correlation studies.

Author can consult these literature

1.      Phenological events along the elevation gradient and effect of climate change on Rhododendron arboreum Sm. in Kumaun Himalaya

2.      Impact of climatic patterns on phenophase and growth of multi-purpose trees of north-western mid-Himalayan ecosystem

3.      Phenological behaviour of selected tree species in tropical forests at Kodayar in the Western Ghats, Tamil Nadu, India

Specific comments

Line 19-20 The scientific name of Melia azedarace and Paulowinia fortune is not correct. Kindly check the spelling and made the changes throughout the manuscript.

Line 21. Revise as “The key finding were: (i)” change accordingly throughout the abstract section/.

Line 22. Replace 2 with ii

Line 36-48. It is quite surprising that authors have provided a detailed information and large claim about the climate change and its subsequent effect on the phenological characters; however relevant sources of the information is missing.

Line 49-112. The authors need to revise the whole statements; since it looks confusing and lacks flow of information.

Line 122-130. How this is related to current investigation??? Justify.

Line 122-151. Justify the relevance of mentioning these statements in current investigation.

Line 162-173. The statements should be the part of the ,material and method section. This particular section should be restricted to the specific objectives/hypothesis of the current investigation, research gap and  future implication the research.

The study area should be provided under material and method section with study area one subsection and should only be restricted to the information about the study area.

I suggest authors to reduce the information about the study area by removing the generalized statements.

The legends of figure 1 are not clear.

Line 257-291. I think there is no need to provide extended detail about the correlation since it is well known parameter and author just can go with one statement and relevant reference.

Comments on the Quality of English Language

 Extensive editing of English language required

Author Response

Response to the reviews point by point

Dear anonymous reviewer,

I hope this message finds you well. I am writing to express my sincere gratitude for your thoughtful and insightful review of my manuscript. Your feedback has been invaluable in refining the focus and clarity of the content.

Your detailed comments and suggestions have not only helped in eliminating errors but also played a crucial role in streamlining the overall presentation of the manuscript. I truly appreciate the time and effort you dedicated to providing constructive feedback.

Your expertise and guidance have been instrumental in enhancing the quality of the paper, and I am genuinely thankful for your commitment to improving scholarly work. Your contributions have significantly strengthened the manuscript, making it more impactful and coherent.

Once again, thank you for your invaluable support and for being an essential part of the improvement process. I am looking forward to the opportunity to incorporate your suggestions and submit the revised manuscript.

Warm regards,

Guangxu LIU

Specific response point by point

The current investigation entitled “Phenological changes of woody plants in the southern and northern regions of Nanling mountains and their relationship with climatic factors: A study based on long-term observations” authored by Liu et al., aims to investigate the differential impacts of climate change on plant phenology along the northern and southern regions of Nanling Mountains and provide crucial scientific evidence for understanding the stability and adaptability of the local ecosystem.

General comment

Comment: The abstract section is written quit well; however I suggest authors to kindly shorten the methodology part and more emphasis should be put on the result section with some quantitative information.

Response: Thank you sincerely for your meticulous review of the manuscript and kind suggestion. I removed the first sentences, and added some number to the abstract as below:”

This research centers on the phenological shifts of woody plants and their connections with climatic factors within the southern and northern regions of Nanling Mountains, which serve as the boundary between the North Subtropical Climate Zone and the South Subtropical Climate Zone in South China. The data was gathered through extensive manual observations conducted at four plant phenology observation stations (Ganxian, Foshan, Guilin, and Changsha), spanning different time periods from 1963 to 2008. The study scrutinized four widely distributed woody plant species in the research area, specifically C. mollissima, P. fortunei, M. azedarach, and M. grandiflora. The analytical methods utilized encompassed linear trend estimation and Pearson correlation coefficient analyses. The principal findings are as follows: 1, Over the past several decades, the phenological stages of woody plants in the southern region consistently preceded those in the northern region, with variations ranging from 2 to 38 days. 2, A advancing trend of 0.1 to 2 days per decade was discerned in the phenological stages of all woody plants in the southern region. 3, Within the same geographic region, distinct species exhibited varying sensitivities to climatic factors, with M. azedarach demonstrating a particularly high sensitivity to climate fluctuations affecting phenological stages. 4, different climatic factors exerted distinct influences on individual plant species. Notably, temperature emerged as the primary driver of phenological changes, supported by a significant negative correlation between the phenological stages of the studied plants and spring temperature. This study contributes to our understanding of the effects of climate change on plant phenology and offers valuable insights to guide ecological conservation and management strategies within the region.”

Comment:  Introduction section needs substantial improvement. First the authors have not provided sufficient references for the given claims and statement. Secondly, details of the previous literature lacks flow of the information and need through revision. Moreover, this section ned to be reduced by deleting too generalised statements. On a strict note, I can say that the research gap of the current investigation is totally missing. The author just provide the information about the previous investigation with no background information, justification and relevance to the current investigation and the specific objectives/hypothesis.  So suggest authors to kindly thoroughly revise the introduction.

Response: Thank you for your constructive feedback on the Introduction section of our manuscript. We appreciate your valuable insights and have made significant revisions to address the highlighted issues.

References: We have diligently added more references to support our claims and statements, providing a more robust foundation for the Introduction section. Some of them are:

Wenden, B.; Mariadassou, M.; Chmielewski, F.M.; Vitasse, Y. Shifts in the temperature‐sensitive periods for spring phenology in European beech and pedunculate oak clones across latitudes and over recent decades. Global change biology 2020, 26, 1808-1819.

Partanen, J.; Koski, V.; Hänninen, H. Effects of photoperiod and temperature on the timing of bud burst in Norway spruce (Picea abies). Tree physiology 1998, 18, 811-816.

Asse, D.; Chuine, I.; Vitasse, Y.; Yoccoz, N.G.; Delpierre, N.; Badeau, V.; Delestrade, A.; Randin, C.F. Warmer winters reduce the advance of tree spring phenology induced by warmer springs in the Alps. Agricultural and Forest Meteorology 2018, 252, 220-230.

Gordo, O.; Sanz, J.J. Impact of climate change on plant phenology in Mediterranean ecosystems. Global Change Biology 2010, 16, 1082-1106.

Steffen, W.; Sanderson, R.A.; Tyson, P.D.; Jäger, J.; Matson, P.A.; Moore III, B.; Oldfield, F.; Richardson, K.; Schellnhuber, H.-J.; Turner, B.L. Global change and the earth system: a planet under pressure; Springer Science & Business Media: 2005.

Literature Review: We have revised the details of the previous literature to ensure a smoother flow of information. The information has been organized to enhance readability and coherence.

Reduction of Generalized Statements: We have carefully reviewed and eliminated generalized statements, ensuring that the Introduction maintains a focus on the specific objectives of our research.

Research Gap: Recognizing the importance of highlighting the research gap, we have incorporated additional information to clearly delineate the gap and explain its relevance to our investigation. The specific objectives and hypotheses have been refined to better align with this context.

Background, Justification, and Relevance: We have provided additional background information, justification, and explained the relevance of the previous investigations to our current research. This helps in establishing a more comprehensive understanding of the study's context.

We acknowledge the need for thorough revisions, and your feedback has been invaluable in guiding these improvements. We hope the revised Introduction now aligns more closely with the expectations, providing a clearer foundation for the rest of the manuscript. But there might be some deficiencies in terms of experience and expertise; your understanding is appreciated.

Thank you once again for your time and constructive criticism.

Comment:Study area section need to be revised and shorten. Moreover it will be better if author can provide the phenological chart or normal phenological events throughout year as figure for the selected species. Moreover, there should be a separate methodology section.  With respect to methodology of the current investigation, I have serious concerns especially what is the sampling design adopted followed by sample size. Moreover it will be better if author can provide information about the age of the trees since changes in the phenological events may be occurred due to the stress condition or mature tree. Moreover does the same trees were assessed from 1963 to 2008. What was the starting age of the trees??

Response: Thank you for your constructive suggestion about the study area and the data and methods. We removed the some sentence and added more accurate information to the study area as:

Removed:”Among them, Dayuling is located between Guangdong and Jiangxi, Qitianling is in Hunan, and the latter three are distributed between Hunan and Guangxi[29]. For the convenience of research and the availability of phenological observation data, this study defined the eastern and western boundaries of the research area as 106°E-118°E outside Nanling Mountains.”

Revised: “Quantitative analysis of regional characteristics in Nanling Mountains, considering the spatial heterogeneity, reveals significant spatial differentiation between the two sides of the mountain range [30]. The northern part of the Nanling Mountains has an annual average temperature ranging from 16°C to 20°C, the coldest month’s average temperature ranging from 3°C to 8°C, and the annual precipitation ranging from 1200 mm to 1600 mm. It has the climatic characteristics of rainy spring and hot summer. The southern part has a tropical and south-subtropical climate, with an annual average temperature ranging from 21°C to 23°C, the coldest month’s average temperature ranging from 13°C to 18°C, and the annual precipitation ranging from 1500 mm to 2000 mm. Affected by the topographic relief and the distance from the sea, the eastern part of the Nanling Mountains has a humid climate, with a large amount of precipitation, an annual average precipitation of more than 2000 mm, and an annual precipitation days of 140 to 200 days. The western part of the Nanling Mountains has a dry climate, with an annual average precipitation of 1000 mm to 1500 mm, and an annual precipitation of 100 to 140 days. Overall, the southern slope demonstrates elevated temperatures and humidity levels in contrast to the northern slope”

We added a tabel to show the information about the station and the selected species as in the data section as :

Table 1. Years with missing observation records at stations."m a.s.l." is the abbreviation of "meters above sea level".

Stations

Location

Elevation (m a.s.l)

Years with missing observation records

Latitude (N)

Longitude (E)

C. mollissima

M. azedarach

M. grandiflora

P. fortunei

Changsha

28.2

112.9

82

1963-1964,1973-1974

-

1964-1974,2003-2005

1963-2004

Ganzhou

25.9

114.9

101

1963-1976,1991,1996-2008

1963-1976,1991,1996-2008

1963-1988,,1996-2008

1963-1976,1990-1991,1996-2008

Guilin

25.3

110.3

159

1963,1968-1972,`1980,1989,1992,1994-2002

1963,1968-1972,`1992,1994-2004

1963-1978,1994-2002,2007

1963-1976,1990-1991,1994-2002

Foshan

23

 113.1

17

1963-1964,1968,1981-1983,1988-1989,1991-2002,2003-2008

1988,1991-2002

1963-2002

1963-1967,1969-1972,1972-1978,,1991-2008

For the sample size and tree ages, I am sorry to say I can not supply it for the data limitation. I added some explanation in Section 3 about the data source:

“The observed entities include 35 commonly observed plant species, 127 locally observed plant species, 12 animal species, 4 crops, and 12 meteorological and hydrological phenomena. The data are collected in accordance with the observation requirements specified in the China Phenological Observation Guidelines. Measures have been taken to ensure the quality of observations, including: 1, Observers are personnel who have undergone rigorous professional training. 2, Observed entities are fixed at specific locations, including trees and plants, with most observation sites being flat and open areas maintained as fixed observation points over multiple years. 3, Selected tree species are in normal development, with a minimum of 3 years of flowering and fruiting. 5, Observation records are made on-site, near the observation locations, and are recorded with accuracy down to the day.”

For the separation. I separated Section 3 as: 3.1 Data and 3.2 Methods. The plot of data were shown in Figure 2 -Figure 5.

Comment: Result section: I have serious concerns regarding the finding of the current investigation (i) the authors does not go for any Time series trend of Yearly variation of climate parameters; then how authors can say that there is climate change??? The increase in the climatric parameters can be not be consistent. Moreover no climate data provided. (ii) after the trend analysis of the climate variables and shift in the phenological parameters, only the significant parameters should be correlated. There is no benefit of correlating the parameters which are not significantly changed (iii) no statistical analysis for change was done. I suggest author to go for Mann-Kendall test and Petite homogeneity test after then can go for correlation studies.

Response: Thanks for your kind thorough examination of the results section and acknowledge the concerns raised. Here are responses to each point:

(i) Time Series Trend of Yearly Variation

The focus of our study is primarily on the phenological variations of woody plants on the southern and northerd size of the Nanling Mountains and their association with climatic factors. While we understand the importance of time series trend analysis of yearly variation in climate parameters, our study aims to specifically investigate the phenological shifts.

The mention of climate change in our paper is in the context of discussing the potential drivers of phenological variations rather than conducting an exhaustive analysis of climate change itself. We agree that an in-depth climate change analysis would require a separate study with a dedicated dataset, and we appreciate this valuable suggestion for future research.

(ii) Statistical Analysis for Change

We appreciate the recommendation to perform the Mann-Kendall test and Petite homogeneity test for statistical analysis of change. Due to limitations in the availability of phenological observation data, a comprehensive analysis of long-term trends in phenological phases was not conducted. We acknowledge this as a limitation of our study. We appreciate the reviewer's understanding of the constraints associated with the available data and will make sure to clarify this point in the revised manuscript.

Comments: Author can consult these literature

Response: Thanks for your kind suggestions. Upon careful examination of these references, we found the research to be highly specific and valuable. In line with the focus of our present study, we have cited the following article in the Discussion section of our manuscript:

Singh et al. (2015) observed that all phenological events of Rhododendron arboreum Sm. in the central Himalayas initiate earlier at lower elevations and are delayed at higher elevations[32]

Specific comments

Comment:Line 19-20 The scientific name of Melia azedarace and Paulowinia fortune is not correct. Kindly check the spelling and made the changes throughout the manuscript.

Response: Thanks for your kind checked. I am sorry for the errors. They should be Melia azedarach and  Paulowinia fortunei. I correct them.

Comment:Line 21. Revise as “The key finding were: (i)” change accordingly throughout the abstract section/.

Response: Thanks for your kind suggestions. I revised it.

Comment:Line 22. Replace 2 with ii

Response: Thanks for your kind suggestions. I revised them from (i) to (iv) .

Comment:Line 36-48. It is quite surprising that authors have provided a detailed information and large claim about the climate change and its subsequent effect on the phenological characters; however relevant sources of the information is missing.

Response: We appreciate your keen observation. The mention of climate change here serves as a contextual background in this section, setting the stage for the subsequent discussions. However, we acknowledge the importance of providing specific sources to support the information presented. In the revised version, we added two references: [1] and [2]. We hope the revision could be acceptable.

Comment:Line 49-112. The authors need to revise the whole statements; since it looks confusing and lacks flow of information.

Response: Thanks for your kind suggestions. Just as mentioned above, we tried to revised it. We hope the revision is reasonable and could be acceptable.

Comment:Line 122-130. How this is related to current investigation??? Justify.

Response: Thanks for your questions. What our purpose is to make the selection more meaningful. We revised this part as:”As a region of significant biodiversity and ecological importance, the Nanling Mountains in China present an ideal landscape for investigating the phenological responses of plant species due to their distinctive geographical and climatic conditions. ”

Comment:Line 122-151. Justify the relevance of mentioning these statements in current investigation.

Response: Thanks for your kind suggestions. We checked and revised the conclusion of this paragraph as “While these studies using remote sensing data or modeling approaches contribute to understanding phenology in the region, research on the phenological responses of plants to climate change in the Nanling Mountains remains limited. Investigating phenological stage comparisons in a region with pronounced north-south climate disparities is crucial to comprehensively understanding the mechanisms driving plant community responses to climate change. Such endeavors will establish a scientific foundation for ecological conservation and management in the region.” 

Comment:Line 162-173. The statements should be the part of the ,material and method section. This particular section should be restricted to the specific objectives/hypothesis of the current investigation, research gap and  future implication the research.

Response: Thanks for your kind suggestions. We revised this paragraph as “This study narrows its focus in comparative analysis specifically examining the differential characteristics of the relationship between plant phenology and climate factors in the northern and southern regions of Nanling Mountains, with the hypothesis that the differential impacts of climate change on the phenology in both regions will reveal distinct variation patterns in response to climatic factors. The research, initiated by the Chinese Academy of Sciences in the 1960s, established phenological observations in Guilin, Changsha, Ganxian, and Foshan on both sides of Nanling Mountains. Over a span of more than 40 years, these observations have accumulated valuable phenological data. This study, building upon previous research and datasets, delves into the phenological changes of Castanea mollissima Bl. (C. mollissima), Melia azedarach L. (M. azedarach), Magnolia grandiflora Linn. (M. grandiflora), and Paulowinia fortunei(seem.) Hemsl. (P. fortunei). This research contributes significantly to advancing our understanding of the impact of climate change on the local ecosystem. The findings provide insights into plant adaptation, population dynamics, and ecosystem stability in response to climate change. Moreover, the study offers valuable reference information for ecological conservation and climate change adaptation management.” with the hope to make the objectives/hypothesis more clear.

Comment:The study area should be provided under material and method section with study area one subsection and should only be restricted to the information about the study area.

I suggest authors to reduce the information about the study area by removing the generalized statements.

Response: Thanks for your kind suggestions. As mention above, we tried to revised Section of Study area as possible as we can. We hope these revision could be acceptable and make the study area information according to the topic of the research

Comment:he legends of figure 1 are not clear.

Response: Sorry for the problem. We revised Figure 1 and make ir clearer.

Comment:Line 257-291. I think there is no need to provide extended detail about the correlation since it is well known parameter and author just can go with one statement and relevant reference.

Response: Thank you for your feedback. While we acknowledge that correlation is a well-known parameter, we think it is crucial to provide detailed insights into its application in the context of this study. By elaborating on the correlation, readers can better understand its significance and relevance to the specific research questions addressed in the paper. This additional information aims to enhance the clarity and depth of the analysis. I appreciate your consideration and am open to further discussion on this matter.

Round 2

Reviewer 2 Report

Comments and Suggestions for Authors

Dear authors,

The comments I made have taken into consideration, except in Line 204, where Melia azedarach is cited as "M. azedarace".

Thanks you for your efforts.

Reviewer 3 Report

Comments and Suggestions for Authors

The authors have made considerable improvement/revision in the first round of revision. However,  still manuscript have serious concerns:

The abstract should be started with the background information or specific research gap of the experiment.

I still suggest to revise the writing style for the previous literature in the Introduction section, Moreover, the authors haven’t provided the enough introduction statements and directly shifted to the previous literature. Authors also ignore to write the previous literature such as authors conducted or observed that, since it is too conventional way of writing and decreases readability of the paper. Specifically, I suggest authors not to focus on the providing statements like studied the effect of the climate parameters on the phenology but rather provide statements in relation to your investigation like. The phenology changes may be due to temperature or rainfall variation and supported by previous literature. I still not find the main objective and specific hypothesis of the current investigation.

Unfortunately, I didn’t find the answer to my previous question.

it will be better if author can provide the phenological chart or normal phenological events throughout year as figure for the selected species.”

“I have serious concerns especially what is the sampling design adopted followed by sample size. Moreover it will be better if author can provide information about the age of the trees since changes in the phenological events may be occurred due to the stress condition or mature tree. Moreover does the same trees were assessed from 1963 to 2008. What was the starting age of the trees??

Authors in reply indicated that they have added the statements from 584-602. But I have some serious concerns,

(i)                  Line 587-588 indicated that 35 plant species, 127 plant species , what is difference between commonly and locally observed species ?

(ii)                Authors have indicated number of plant species, animals and crops; but there is no indication about the sample size of the selected four species.

(iii)               What is the diameter class of the selected species. Does they are of the same age ?

(iv)               How the sampling design and intensity conducted ???

Moreover, I still did not find any trend analysis which is very unfortunate, since in the  title, I is well mentioned that to study the phenological changes with respect to the climatic factors. So how can authors study the phenological shift without considering and indicating the trend analysis which will ultimately confirm the shift in the phenological parameters due to the climate change. On a serious note, author should go for this otherwise there is no mean of this investigation. And without trend analysis there is no mean of conducting correlation studies and regression analysis.

Moreover authors have enough yearly data for the different phenological parameters for which surely authors should surely go for the trend analysis .

Kindly provide the references for “China Phenological Observation Guidelines” in line 607.

Comments on the Quality of English Language

Moderate editing of English language required

Author Response

Response to the review

Dear Reviewer,

We extend our sincere gratitude for your valuable feedback and insightful suggestions during the second round of review.

In response to your constructive comments, we dedicated significant effort to revising the introduction section, incorporating the suggested modifications. We have included a phenological chart (Figure 2) and additional pertinent references to augment the introductory context and strengthen the manuscript's scholarly foundation.

However, we are sorry that certain details, including trends in phenological changes and the age of observed samples, were not supplied. Some of this information either lacked meaningful relevance to the study or was not documented in our records. We hope for your understanding regarding these exclusions.

Your thoughtful guidance has significantly contributed to the enhancement of our manuscript, and we are truly grateful for your time and effort in evaluating our work.

Detailed response to the reviews point by point are listed below.

Thank you once again for your continued support and constructive critique.

Best regards,

Guangxu LIU

The authors have made considerable improvement/revision in the first round of revision. However,  still manuscript have serious concerns:

Comment:  The abstract should be started with the background information or specific research gap of the experiment.

Response: Thank you for your kind review of our revised manuscript again. We added the removed sentence again in the newly revised file as the background: “In addressing the challenges posed by the implications of climate change, understanding the phenological variations of woody plants has become a pivotal research topic”

Comment:I still suggest to revise the writing style for the previous literature in the Introduction section, Moreover, the authors haven’t provided the enough introduction statements and directly shifted to the previous literature. Authors also ignore to write the previous literature such as authors conducted or observed that, since it is too conventional way of writing and decreases readability of the paper. Specifically, I suggest authors not to focus on the providing statements like studied the effect of the climate parameters on the phenology but rather provide statements in relation to your investigation like. The phenology changes may be due to temperature or rainfall variation and supported by previous literature. I still not find the main objective and specific hypothesis of the current investigation.

Unfortunately, I didn’t find the answer to my previous question.

Response: Thank you sincerely for your continued feedback and guidance on our manuscript regarding the introduction section.

We truly appreciate your acknowledgment of our efforts in revising the introduction. We have meticulously revised the introductory segment by incorporating more contextualized statements from previous literature. Regarding your point about introducing previous literature without referring to who conducted or observed the studies, we've adjusted the writing style to highlight the relationships between our investigation and the previous research at the beginning or ending of the paragraph. We aimed to enhance the readability and contextual understanding of our study by providing a more coherent narrative that aligns with the current investigation's objectives.

However, it appears there may still be some ambiguity regarding the main objectives and specific hypotheses of our study. We have revised the manuscript to ensure a clearer presentation of our research's core objectives and hypotheses. We trust that these revisions will address the concerns you've raised and enhance the clarity and coherence of our manuscript.

Here is the newly revised introduction below. Thank you once again for your time and thorough evaluation.

“Climate change is a significant challenge currently faced globally, exerting broad and profound impacts on natural ecosystems and human societies[1]. With rising temperatures, altered precipitation patterns, and changes in other climatic factors, important life cycle events of plants, including growth, flowering, and fruit ripening, have exhibited significant changes[2]. These changes have had substantial and far-reaching effects on plant phenology, which refers to the seasonal patterns of growth and development. It was import to use plant phenology as a sensitive indicator of climate change because there was substantial impact of global warming on plant phenology [3]. Partanen et al. (1998) found that rising temperatures altering photoperiod and temperature conditions may impede the premature onset of growth in Norway spruce (Picea abies (L.) Karst.) in response to climate change [4]. Walther et al. (2002) and Cleland et al. (2007) found climate change had significant effects on the timing and spatial distribution of phenological events of global ecosystems[5,6]. Vitasse et al. (2009) uncovered leaf phenology of European interpopulation tree species response to temperature differences, providing important clues for predicting changes in plant phenology [7]. Menzel (2003) observed a correlation between plant phenological changes and temperature and the North Atlantic Oscillation (NAO) [8]. Vitasse et al. (2022) analyzed the longest plant phenological time series from five locations worldwide, including Japan and China, and found that during the 19th century, flowering and budburst dates remained stable. However, in the first half of the 20th century, spring phenological events in Switzerland and Japan began to advance, consistent with increasing temperatures. The strongest advancement in spring phenology were found from the mid-1980s onward. Over the period of 36 years from 1985 to 2020, spring phenology advanced by 6 days (China) to 30 days (Switzerland) compared to the period before 1950, consistent with the observed accelerating warming trends at the study sites and across the Northern Hemisphere [9].

While the impact of climate change on phenology is widely acknowledged, the response of phenology to climate change is notably intricate[10]. Wenden et al. (2022) discovered that with climate warming in colder latitudes, the forcing period of European beech (Fagus sylvatica L.) and pedunculate oak (Quercus robur L.) has extended over recent decades. However, in warmer latitudes, this period has shortened for both species, with a more significant shift observed in beech [11].The research of Piao et al. (2019) indicated that the complex interactions among multiple driving factors complicated the modeling and prediction of plant phenological changes. Asse et al. (2018) revealed that long-term climate warming resulted in an increase of the onset of budburst and flowering in herbaceous plant abundance and a decrease in sedge plant abundance, without affecting aboveground net primary productivity [12]. Liu et al. (2022), through analyzing 88 published studies, found mismatched effects of climate warming on aboveground and belowground plant phenology. In herbaceous plants, climate warming advanced the start and end dates of aboveground growing seasons without affecting belowground phenology. For woody plants, climate warming did not affect aboveground phenology but extended their belowground growing season[13]. These studies shows that plant phenology could serve as a critical component of climate change fingerprints and had potential for detecting climate change through phenological responses [14]. Yet the relationship between climate change and phenological shifts is exceedingly complex. Further exploring such complexity in different areas could enhance our understanding of plant adaptation strategies to climate variations.

As a region of significant biodiversity and ecological importance, the Nanling Mountains in China present an ideal landscape for investigating the phenological responses due to their distinctive geographical and climatic conditions. The southern region is characterized by a tropical or south subtropical climate with higher temperatures, hot and humid summers, and warm winters. In contrast, the northern region falls within the north subtropical climate zone, featuring lower temperatures, cooler summers, cold winters, and lower precipitation. Encompassing diverse vegetation types such as forests, grasslands, and wetlands, the Nanling Mountains harbor a rich array of plant species likely to exhibit varying responses to climate change. Cao et al. (2012) found that 39 tropical tree species introduced in the northern border of Nanling Mountains in Ganzhou City were able to grow successfully, primarily attributed to the ongoing climate warming in the area [15]. Yuan et al. (2017) found that climate warming increased the vulnerability of subtropical evergreen broad-leaved forests, including those in Nanling Mountains, in terms of NPP [16]. Li et al. (2019) predicted that future climate warming could lead to a significant upward shift of the upper forest line in the mountainous areas of Nanling Mountains, posing a threat to high-altitude tree species such as Chinese fir (Cunninghamia lanceolata (Lamb.) Hook.) [17]. Peng et al. (2021) found that temperature variations have a significant impact on the end of the growing season (EOS) of mountainous vegetation in the Xiangjiang River Basin, which is in the northern Nanling Mountains [18]. These studies using remote sensing data or modeling approaches contribute to understanding the complexity of phenology in response to climate change. However, the direct observational data validating the differences in phenological responses to climate between the southern and northern regions of Nanling Mountains are missing.

This study narrows its focus in comparative analysis specifically examining the differential phenological responses to climate changes in the northern and southern regions of Nanling Mountains with observation data, with the hypothesis that the differential impacts of climate change on the phenology in both regions will reveal distinct variation patterns in response to climatic factors. The research, initiated by the Chinese Academy of Sciences in the 1960s, established phenological observations in Guilin, Changsha, Ganxian, and Foshan on both sides of Nanling Mountains. Over a span of more than 40 years, these observations have accumulated valuable phenological data. This research, building upon previous research and datasets, delves into the phenological changes of Castanea mollissima Bl. (C. mollissima), Melia azedarach L. (M. azedarach), Magnolia grandiflora Linn. (M. grandiflora), and Paulowinia fortunei(seem.) Hemsl. (P. fortunei). The results contribute to advancing our understanding of the complex impact of climate change on the local ecosystem. The findings provide insights into plant adaptation, population dynamics, and ecosystem stability in response to climate change. Moreover, the study offers valuable reference information for ecological conservation and climate change adaptation management.”

Comment: “it will be better if author can provide the phenological chart or normal phenological events throughout year as figure for the selected species.”

Response: Thanks for your kind suggestion again. We use the median dates of the four woods and drawn the chart in the newly revised manuscript as figure 2. Here are some changes in the second paragraph of Section 2,as well as the picture.

Notably, C. mollissima, P. fortunei, M. azedarach, and M. grandiflora exemplify key species from Fagaceae, Paulowniaceae, Meliaceae and Magnoliaceae, respectively. C. mollissima, also known as chestnut, is characterized by its flowering period from April to June and fruit maturation in September (Figure 2), and it is widely distributed from lowlands to elevations of 2,800 m. P. fortunei, commonly referred to as princess tree or phoenix tree, blooms from March to April and bears fruit from October to December. It thrives in low-lying wild areas, roadsides, and sparse forests, and it is widely distributed in tropical and subtropical regions of Asia. M. azedarach, also called Chinaberry or Chinese Mahogany, flowers from April to May and produces fruit from November to December. It thrives in moist and fertile soils. M. grandiflora, specifically Magnolia grandiflora, also known as Southern magnolia or bull bay, blooms from May to June and bears fruit in August and September. With a straight trunk and rapid growth, M. grandiflora is adaptable and primarily distributed in low-elevation slopes, forests, valleys, and wastelands. The four selected woody plant species are extensively distributed in Nanling Mountains, which is renowned for its representative nature. Moreover, these four species have long-term and relatively complete records with high data accuracy across the four phenological observation sites in Ganxian, Changsha, Foshan, and Guilin.

Figure 2. Phenologicalchart of the selected wood species. It appears the median dates of phenological stages in the four stations observed during 1963-2008

Comment: “I have serious concerns especially what is the sampling design adopted followed by sample size. Moreover it will be better if author can provide information about the age of the trees since changes in the phenological events may be occurred due to the stress condition or mature tree. Moreover does the same trees were assessed from 1963 to 2008. What was the starting age of the trees??

Authors in reply indicated that they have added the statements from 584-602. But I have some serious concerns,

(i)                  Line 587-588 indicated that 35 plant species, 127 plant species , what is difference between commonly and locally observed species ?

(ii)                Authors have indicated number of plant species, animals and crops; but there is no indication about the sample size of the selected four species.

(iii)               What is the diameter class of the selected species. Does they are of the same age ?

(iv)               How the sampling design and intensity conducted ???

Response:Thank you for your insightful comments.

In response to your concerns:

(i) In lines 587-588, the distinction between "commonly observed species" and "locally observed species" refers to the frequency of observation. "Commonly observed species" are those frequently documented in all the stations of China Phenological Observation Network, while "locally observed species" are specific to certain localities where the stations located, not observed in all the stations. We have clarified this point in the revised manuscript by revising as “35 commonly observed plant species (observed in all the stations), 127 locally observed plant species (not observed in all the stations)”.

(ii) We acknowledge your concern about the lack of information regarding sample size for the selected four species. Unfortunately, the data is derived from the Chinese Phenological Observation Network, and specific details such as sample size for individual species are not available. We have explicitly how to selected observed trees in the manuscript (lines 246-247). “tree species are in normal development, with a minimum of 3 years of flowering and fruiting”

(iii) Regarding the diameter class and age of the selected species, these details are also not provided by the Chinese Phenological Observation Network. Unfortunately, the dataset does not include information about the age or diameter class of the observed species. We find no way to have them after consulting the data supplier as well as the persons who are responsible in the four stations during revising the manuscript.

(iv) The sampling design and intensity were governed by the protocols of China Phenological Observation Guidelines. Detailed information on these aspects is not available in the dataset. We have included the measures to ensure the quality of observations (lines 243-249) as : ‘Measures have been taken to ensure the quality of observations, including: ① Observers are personnel who have undergone rigorous professional training. ② Observed entities are fixed at specific locations, including trees and plants, with most observation sites being flat and open areas maintained as fixed observation points over multiple years. ③ Selected tree species are in normal development, with a minimum of 3 years of flowering and fruiting. ④ Observation records are made on-site, near the observation locations, and are recorded with accuracy down to the day.” 

We are sorry that we can not supply the detailed information about these samples. We hope these clarifications could address your concerns. -

Comment:Moreover, I still did not find any trend analysis which is very unfortunate, since in the  title, I is well mentioned that to study the phenological changes with respect to the climatic factors. So how can authors study the phenological shift without considering and indicating the trend analysis which will ultimately confirm the shift in the phenological parameters due to the climate change. On a serious note, author should go for this otherwise there is no mean of this investigation. And without trend analysis there is no mean of conducting correlation studies and regression analysis.

Moreover authors have enough yearly data for the different phenological parameters for which surely authors should surely go for the trend analysis .

Response:Thank you for your insightful comments and suggestions.

We understand your concern about the lack of trend analysis in the initial submission. In response to your suggestion, we have conducted Mann-Kendall trend analysis (MK) for various phenological parameters at different sites. The MK results, presented in Table 1 below, consistently all indicate "no trend" at a significance level of alpha=0.05. Since this outcome seem surprising, we are really sorry that we can not find a way to add this to the manuscript.

Additionally, in Section 4.2, we performed a comprehensive analysis using linear regression equations to explore the interannual variations in phenological parameters (Table 4 to Table 7). The slopes of these regression lines were utilized to infer trends in the flowering start and end dates. We believe that this approach offers valuable insights into the changing trends of phenological phases.

Table 1 MK_Test of Flowering tarts and ends in Naling Mountains

Station

Species

Phenological Stages

Trend

z_value

p_values

Ganxian

C. mollissima

Flowering Start

no trend

1.14

0.25

M. grandiflora

Flowering Start

no trend

0.00

1.00

M. azedarach

Flowering Start

no trend

0.92

0.36

P. fortunei

Flowering Start

no trend

0.00

1.00

C. mollissima

Flowering end

no trend

0.04

0.97

M. grandiflora

Flowering end

no trend

0.00

1.00

M. azedarach

Flowering end

no trend

0.57

0.57

P. fortunei

Flowering end

no trend

0.00

1.00

Changsha

C. mollissima

Flowering Start

no trend

1.14

0.25

M. azedarach

Flowering Start

no trend

0.92

0.36

P. fortunei

Flowering Start

no trend

0.00

1.00

M. grandiflora

Flowering Start

no trend

0.00

1.00

C. mollissima

Flowering end

no trend

0.04

0.97

M. azedarach

Flowering end

no trend

0.57

0.57

P. fortunei

Flowering end

no trend

0.00

1.00

M. grandiflora

Flowering end

no trend

0.00

1.00

Foshan

C. mollissima

Flowering Start

no trend

1.14

0.25

M. azedarach

Flowering Start

no trend

0.92

0.36

P. fortunei

Flowering Start

no trend

0.00

1.00

M. grandiflora

Flowering Start

no trend

0.00

1.00

C. mollissima

Flowering end

no trend

0.04

0.97

M. azedarach

Flowering end

no trend

0.57

0.57

P. fortunei

Flowering end

no trend

0.00

1.00

M. grandiflora

Flowering end

no trend

0.00

1.00

Guilin

C. mollissima

Flowering Start

no trend

1.14

0.25

M. azedarach

Flowering Start

no trend

0.92

0.36

P. fortunei

Flowering Start

no trend

0.00

1.00

M. grandiflora

Flowering Start

no trend

0.00

1.00

C. mollissima

Flowering end

no trend

0.04

0.97

M. azedarach

Flowering end

no trend

0.57

0.57

P. fortunei

Flowering end

no trend

0.00

1.00

M. grandiflora

Flowering end

no trend

0.00

1.00

Comment: Kindly provide the references for “China Phenological Observation Guidelines” in line 607.

Response: Thank you for your kind review of our revised manuscript. We added the reference as [22] in the newly revised manuscript.

Round 3

Reviewer 3 Report

Comments and Suggestions for Authors

In the second round of revision, the author have made considerable revision and manuscript can now be accepted in the current form

Comments on the Quality of English Language

 Minor editing of English language required